# Single-shot two-dimensional nano-size mapping of fluorescent molecules by ultrafast polarization anisotropy imaging

Peng Wang[1,5], Yogeshwar Nath Mishra[1,2,3,5], Florian J. Bauer [4,5], Murthy S. Gudipati [2] & Lihong V. Wang [1]✉

Molecular size plays a crucial role in determining the physical, chemical, and biological properties of substances. However, traditional fluorescence polarization anisotropy methods struggle to capture fast transient processes or provide plane-specific details. To overcome these limitations, we introduce Compressed Ultrafast Planar Polarization Anisotropy Imaging (CUP2AI). This approach combines femtosecond laser-sheet illumination, molecular rotational diffusivity, and compressed sensing to enable real-time, non-invasive, wide-field anisotropy measurements in both liquid and gas phases. CUP2AI captures 2D molecular size mapping in a single acquisition, granting unprecedented insights into dynamic events across various excitation modes (i.e. both one- and two-photon) and environmental conditions. It enables mapping of molecular volume (500 Å³–80,000 Å³) and hydrodynamic diameter (10 Å–50 Å) based on anisotropy lifetimes. We imaged fluorescein-conjugated dextran in water and polycyclic aromatic hydrocarbons in flames. CUP2AI holds transformative potential for applications ranging from molecular biology and drug design to nanoparticle formation.

Polarization, an intrinsic property of light, has fascinated scientists since its discovery in the 17th century. Beyond its role in revealing the structural organization of matter at the molecular level, polarization serves as a fundamental degree of freedom alongside amplitude and phase in shaping light's properties[1]. The study of polarization anisotropy (PA) of fluorescence from molecules, which involves observing alterations in light polarization during interactions with samples, offers invaluable insights into molecular orientation and alignment[2]. Brownian motion provides direct information on molecular complexes in biology and nanomedicine, such as protein oligomerization[3,4], drug binding[5], nanoscale diffusivity[6], nuclear pore complex sites[7], cytoskeletal protein interactions[8,9], cell cluster size[10], and solvent interactions[11,12]. In environmental and combustion sciences, PA has

been found suitable for nano-sizing harmful and carcinogenic clusters of polycyclic aromatic hydrocarbons (PAHs)[13,14]. This technique was reported for either steady state sample condition[9] or as a time resolved approach[12], with the latter being the preferred choice due to greater sensitivity and higher throughput.

From fluorescence lifetime measurements due to one-photon (1 P) or two-photon (2 P) excitation, time-resolved PA, i.e., the molecular orientation dynamics over a given time window, typically in the range of a few nanoseconds for frequently used fluorophores, can be inferred. Two-photon time-resolved PA has gained popularity because it offers inherent optical sectioning, akin to confocal imaging[5,15]. Further, when working with living cells, the phototoxicity of the unwanted "out-of-focus" near-infrared (NIR) illumination is generally lower

[1]Caltech Optical Imaging Laboratory, Andrew and Peggy Cheng Department of Medical Engineering, Department of Electrical Engineering, California Institute of Technology, 1200 East California Boulevard, Mail Code 138-78, Pasadena, CA 91125, USA. [2]Science Division, NASA-Jet Propulsion Laboratory, California Institute of Technology, 4800 Oak Grove Drive, Pasadena, CA 91109, USA. [3]Department of Physics, Indian Institute of Technology Jodhpur, Jodhpur 342030 Rajasthan, India. [4]Lehrstuhl für Technische Thermodynamik (LTT) and Erlangen Graduate School in Advanced Optical Technologies (SAOT), Universität Erlangen-Nürnberg, Erlangen 91058, Germany. [5]These authors contributed equally: Peng Wang, Yogeshwar Nath Mishra, Florian J. Bauer. ✉e-mail: LVW@caltech.edu

compared to 1 P excitation. And finally, 2 P excitation allows for the elimination of scattered light from the observed fluorophores.

The majority of time-resolved PA data captured within the nanosecond-scale time window are averaged over multiple acquisitions and measures only 0D (single point in space). The detected signal is mostly integrated along the line-of-sight; therefore, they miss plane-specific details. Further, the use of several consecutive laser exposures is known to induce photobleaching of fluorescent markers, impeding reliable measurements. Finally, time-resolved PA solutions for steady-state conditions like microscopy are widespread and available, but the diagnostic developments for transient processes (e.g., flames or sprays) become increasingly more challenging, especially with the demand for process-integrated instantaneous in situ detection.

Traditionally, researchers utilize scanning electron microscopy (SEM) or transmission electron microscopy (TEM) for measuring molecule sizes[16]. Nonetheless, these methods necessitate invasive physical sectioning, resulting in ex situ measurements, requiring highly controlled environments, and being slow and costly. In stark contrast, optical methods using non-ionizing radiation offers a non-invasive, rapid, and cost-effective approach[17]. However, most optical microscopy techniques are limited by diffraction. Successful but expensive super-resolution innovations can only enhance spatial resolutions to the 10–50 nm range[18–20]. Commercial particle sizing instruments based upon diffraction and back-scattering detect particles down to 10 nm. They all unfortunately fall short to resolve typical molecules, which are usually several to tens of angstroms (Å) in size. The solution lies in indirectly extracting information about molecules based on their interaction with photons, which leaves discernible signals. Dynamic light scattering, which exploits temporal auto-correlation function of the light signals scattered from nanoparticles undergoing Brownian motion, measures particle sizes ranging from a few micrometers down to sub-nanometers[21]. However, it is limited to single-point measurements without spatial resolution capability and typically takes seconds to minutes. Polarization, linked to molecule orientation, emerges as an alternative but pivotal indicator to exploit.

Here, we introduce CUP2AI (compressed ultrafast planar polarization anisotropy imaging), a method for capturing picosecond-resolution wide-field anisotropy transients in both liquid and gas phases in a planar (2D) configuration using single-pulse excitation. Our ultrafast technique builds upon the foundations of the world's fastest single-shot compressed ultrafast photography (CUP)[22]. Over the past decade, CUP has recorded various fast phenomena in real-time, such as photon packet propagation[22], photonic Mach cone[23], soliton evolution[24,25], nonlinear light-matter interactions[26,27], plasma formation[28], internodal currents in axons[29], and combustion[30], achieving imaging speeds from billions to trillions of frames per second (fps).

CUP2AI relies on measuring fluorescence photons polarized in directions perpendicular and parallel to the incoming excitation polarization[31]. With the unprecedented time-resolving capability, it becomes possible to observe rapid rotational motions of nano-sized objects (i.e. sub-nanosecond) faster than the fluorescence lifetime. The rotational characteristics of molecules are essentially determined by their sizes and environments, quantitatively described by the Stokes-Einstein-Debye model[2]. Hence, it is straightforward to translate molecular depolarization (orientational diffusivity) rate to molecular size. In this work, we report 2D molecule size mapping in a single-shot across diverse examples, spanning from liquid to gaseous environments and from one-photon to two-photon excitations. Our results demonstrate excellent agreement to previous studies for the flame measurements[32] and overall agreement to the molecular sizes of dextran molecules[31].

## Results and discussion
### System and principle of CUP2AI
A simplified illustration of the system with 1 P excitation is given in Fig. 1. A detailed schematic for both 1 P and 2 P excitations can be found in Fig. S1 in Supplementary Information. All the used components, including manufacturers and models, are listed in Table S1 in Supplementary Information. A Ti:Sapphire femtosecond laser that primarily outputs 800 nm 70 fs pulses at 500 Hz repetition rate was used as the laser source. The laser beam was sent through a nonlinear crystal (BBO) to generate 400 nm second harmonic (SH). The 400 nm SH and the residual 800 nm pulses are utilized to excite 1 P and 2 P fluorescence (1PF and 2PF), respectively, selected by two interchangeable short-pass and long-pass spectral filters (SPF and LPF). The laser beam has a Gaussian intensity profile of $1/e^2$ diameter of about 10 mm and passes through a long-focal-length cylindrical lens (CyL) to form a laser sheet for intersecting a thin $x$-$y$ plane inside the sample (see Fig. 1b). After the cylindrical lens (CyL), a half-wave plate (HWP) and a linear polarizer (P0) adjust the laser fluence while maintaining a linear polarization along the $y$-axis (blue arrow in Fig. 1a). The size of the laser sheet is 5.50 mm × 0.06 mm ($y \times z$). A comprehensive characterization of both 1 P and 2 P laser sheets is provided in Figs. S2, S3, and Table S2 in Supplementary Information.

To detect the emitted optical signal in the $z$-direction, a band-pass filter (BPF) allows the fluorescence spectrum to transmit while rejecting the residual excitation wavelengths. The transmission spectra of the two BPFs employed are plotted in Fig. S4 in Supplementary Information. The intermediate image, formed by the imaging lens assembly (ImL), is split into two copies via a beam splitter (BS): one copy is reflected to an external CCD camera that captures a conventional time-unsheared image of the transient scene, and the other transmitted copy is further relayed onto a digital micromirror device (DMD) for spatial encoding. A computer-generated static pseudo-random binary pattern is displayed on the DMD, which reflects two beam paths, each masked by complementary encoding patterns (see $C_1$ and $C_2$ in Fig. 1c). A high-numerical-aperture stereoscopic lens (SL) used for imaging has dual functions: projecting the image to the DMD and collecting the two masked beams reflected from the DMD. Two relay lenses (ReL1 and ReL2) further send the two encoded images to the entrance of a streak camera. For anisotropy sensing, two polarizers (P1 and P2) are introduced in the two imaging paths, and they are cross-polarized to each other (see the green arrows in Fig. 1d).

Inside the streak camera, the ultrafast image sequence is temporally sheared in the $y$-axis (vertical) in the form of photoelectrons, driven by a fast linear sweeping voltage. Hence, an image frame from time $t + \Delta t$ is displaced downward by $\Delta P$ pixels relative to the frame at time $t$. Here, $\Delta P = (v\Delta t)/d$, in which $v$ represents the shearing speed of the applied voltage ramp and $d$ is the streak camera's pixel size. A stack of spatially encoded, temporally sheared images is time-integrated and recorded by an internal CMOS camera. The streak camera's working principle is described in Methods and Fig. S5a in Supplementary Information. It's worth noting that the imaging speed of CUP2AI is ultimately determined and can be flexibly tuned over a wide range (from 1 million fps to 2 trillion fps) by the sweeping speed $v$. In this work, we opted to image at 125, 50, 25 and 12.5 billion fps (Gfps). The temporal responses and resolutions of different imaging speeds are experimentally characterized and summarized in Fig. S6 and Table S3 in Supplementary Information.

The imaging pipeline results in two simultaneous acquisitions: the time-unsheared view ($E_0$) from the traditional CCD camera and the time-sheared view from the streak camera. The latter one contains two side-by-side images: $E_1$ with P1 and $E_2$ with P2. The mathematical and visual representations of CUP2AI's image formation process are included in Methods and Fig. S5b in Supplementary Information. Compressing high-dimensional data cubes [i.e. fluorescence emission, $I^\parallel(x, y, t)$ and $I^\perp(x, y, t)$ for $y$- and $x$-polarized light] to only three 2D images received by planar sensors ($E_0$, $E_1$, $E_2$) is the key concept underlying CUP2AI, enabling its single-shot imaging capabilities. However, the inverse process of recovering the high-dimensional data from 2D raw images is an ill-conditioned minimization problem, which

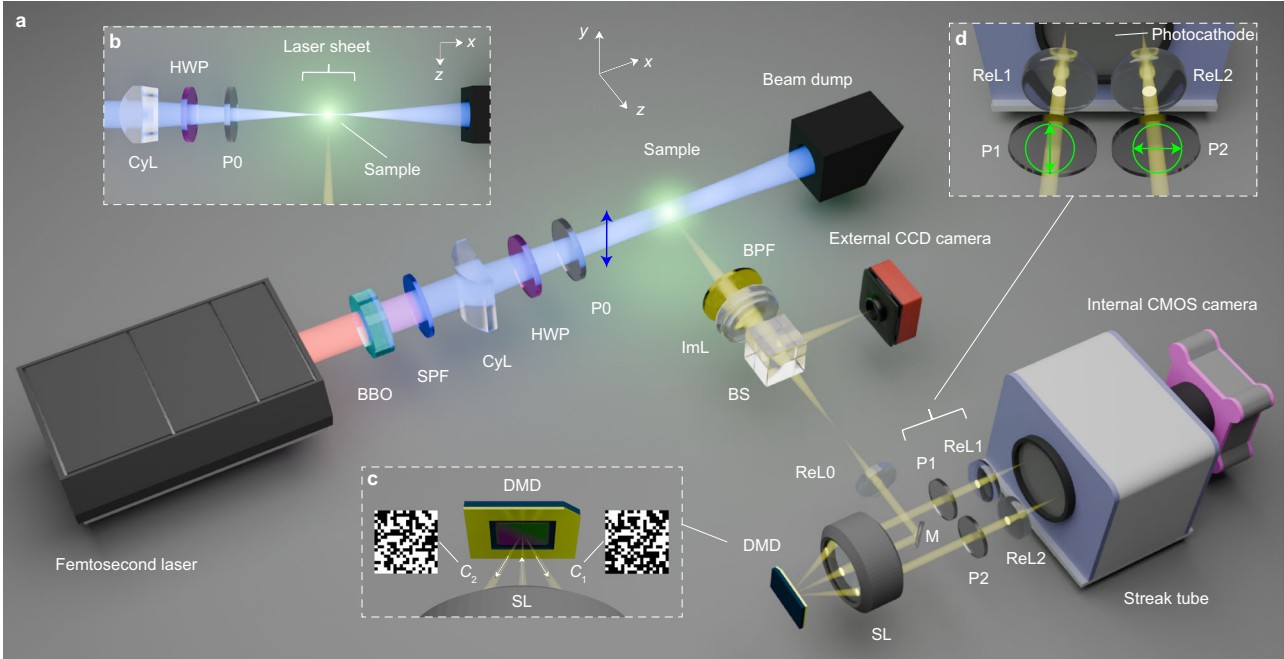

**Fig. 1 | Compressed ultrafast planar-polarization anisotropy imaging (CUP2AI).**
**a** Simplified schematic of the CUP2AI system. BBO beta barium borate, BPF band-pass filter, BS beam splitter, CyL cylindrical lens, DMD digital micro-mirror device, HWP half-wave plate, ImL imaging lens, M mirror, P (P0, P1, and P2): linear polarizers, ReL (ReL0, ReL1, and ReL2): relay lenses, SL: stereoscopic lens, SPF: short-pass filter. The red and blue beams represent the original femtosecond laser light and its second harmonic, respectively. The yellow beam represents the imaging optical path. The polarization axes of the polarizers are represented by the double-headed arrows. The excitation laser pulse is polarized along the vertical direction (*y*). A more detailed system design is illustrated in Supplementary Fig. 1. **b** Top-view of the laser sheet exciting a thin section in the *x-y* plane inside a fluorescent sample. **c** DMD reflects two beam paths from the original imaging beam from SL. The two reflected beams are complementarily coded ($C_1$ and $C_2$) and collected by SL. **d** Close-up view of the system in front of the streak camera's photocathode. The polarization angles of the polarizers P1 and P2 are oriented along the vertical (*y*) and horizontal (*x*) directions, respectively.

requires regularization-based iterative optimization approaches (see Methods for detailed descriptions). It is necessary to mention that only sparse data can be successfully compressed and reconstructed without loss of fidelity. Such a presumption usually holds valid for physically occurring events in the real world. Our previous study showed a typical sparsity of ~95% for most ultrafast phenomena[26]. In this study, we employed pseudo-random masks that generate a well-conditioned sensing matrix, often satisfying the restricted isometry property, enabling accurate recovery of sparse samples. Secondly, the regularization term enforcing sparsity plays a crucial role in guiding convergence.

The detection of fluorescence signals at both polarizations ($I^\parallel$ and $I^\perp$) after short (fs) and polarization-defined excitation can be used to investigate the rotational motion of nano- and sub-nano-sized molecules or particles. Smaller objects show a faster Brownian rotation than larger structures and therefore have shorter polarization anisotropy decay times. Therefore, time resolved measurements of PA can be utilized to gauge the molecule or object size when the environment parameters such as viscosity are known. The equations needed to determinate the anisotropy decay lifetime and ultimately the molecular volume *V* are given in Eqs. (8)–(18) in Methods. Note that to account for the finite temporal resolutions of the CUP2AI system, its temporal point-spread-functions were characterized, and afterwards a numerical method used to approximate a deconvolution was implemented to derive PA lifetimes with a higher accuracy (see Section 5, Figs. S6–S8 and Table S3 in Supplementary Information for details of this deconvolution approach). To convey a clear overview of our data processing streamline, we draw a flow chart in Fig. S9 in Supplementary Information, including all major steps. A package containing example data and the processing code is publicly available in Code Ocean. It contains the major steps of obtaining molecule size maps from intensity movies.

## Molecule sizing using 125-Gfps PA imaging of 1PF in a liquid environment

Liquid is where most chemical reactions and biological processes take place. The characteristics of these reactions and processes heavily depend on the sizes of the molecules involved. Therefore, mapping the molecule size distribution over space is indispensable in biological, medical, and chemical studies. Using fluorescein-based molecules dispersed in water, we demonstrated single-shot wide-field 2D mapping of molecular size based on 1PF anisotropy. Fluorescein is an essential fluorescent chemical with wide biomedical applications such as ophthalmological diagnosis and microscopy imaging. Its derivative, fluorescein isothiocyanate (FITC), when conjugated to groups of dextran (FITC-dextran), is extensively employed in microfluorimetry to study microcirculation, cell permeability, and cell division.

In our experiments, we investigated three aqueous samples of molecules with different sizes: pure fluorescein dye (FL), FITC-dextran with molecular weight (MW) of 4000 Da (4 K), and FITC-dextran with MW of 20,000 Da (20 K). They all have a concentration of 50 μM. Surface-polished cuvettes, filled with liquid samples, are used for imaging. Details about the chemicals and the sample preparation procedure can be found in Methods and Table S4 in Supplementary Information. Here, we assume a homogeneous distribution of the molecules in each cuvette and a transmission mask of the Caltech logo was placed on the detection side of the cuvette to introduce spatial features. To further test the technique with different molecules in the detection volume, two cuvettes with two masks of the Caltech lettering were placed next to each other and excited by the same laser pulse. Schematics of the single- and dual-cuvette configurations are drawn in Fig. 2a, b, respectively. The dual-cuvette imaging additionally highlights the large dynamics that our technique can accommodate as multiple diversely sized molecules can be resolved in a single shot. The 400-nm ultra-violet (UV) laser sheet,

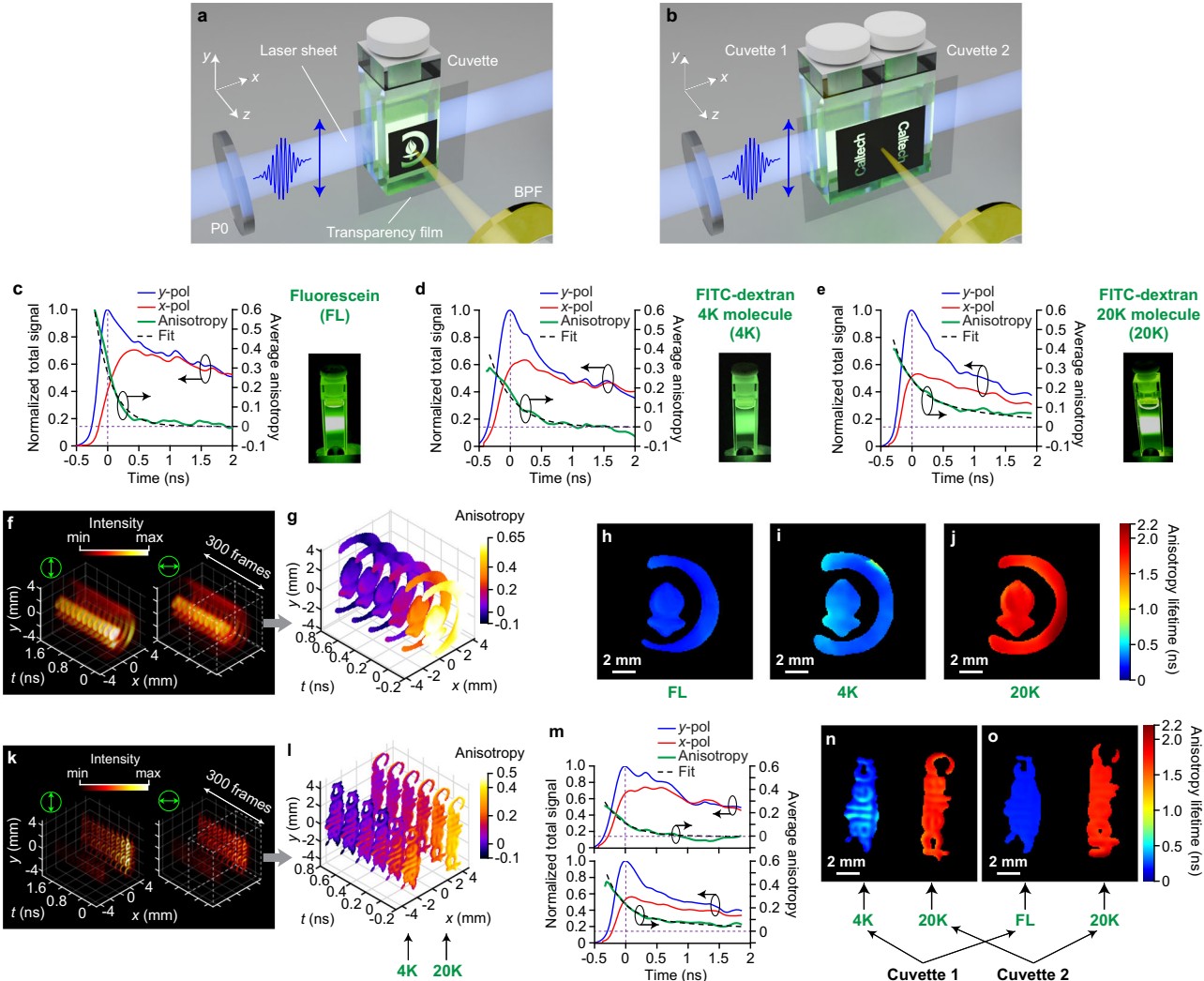

**Fig. 2 | 125-Gfps polarization anisotropy imaging of fluorescein, 4 K, 20 K molecules and their combinations, excited by a single UV femtosecond laser pulse via one-photon process. a** Schematic of single cuvette with one type of fluorescence molecule and single pattern. P0: linear polarizer, BPF: band-pass filter. The excitation pulse is polarized vertically along the *y*-direction. Spatial pattern is printed on a transparency film and applied on the cuvette's front. **b** Schematic of dual cuvettes side-by-side with two types of fluorescence molecules and two patterns. In (**a**, **b**) the blue and yellow beams represent the fluorescence excitation and emission, respectively. **c**–**e** Left *y*-axis: evolutions of normalized spatially integrated intensities from both *x*- and *y*-polarization channels (red and blue solid lines). Right *y*-axis: evolutions of spatially averaged anisotropy. The green solid lines are the measured anisotropies, and the black dashed lines represent exponential fits. Right insets: photographs of excited fluorescence samples in cuvettes, captured using the same exposure parameters. The inset photos show the greenish image of fluorescence of 4 K, while the whiteish images of FL and 20 K are due to higher intensity under the same exposure condition. **f**–**j** Exemplary intensity, anisotropy,

and lifetime results of single-cuvette experiments. **k**–**o** Exemplary intensity, anisotropy, and lifetime results of dual-cuvette experiments. **f**, **k** Reconstructed intensity evolutions of one-photon fluorescence from (**f**) fluorescein molecule in the single-cuvette configuration and (**k**) 4 K and 20 K molecules in the dual-cuvette configuration. 12 exemplary snapshots are selected among a total of 300 frames. The green double-headed arrows represent the orientations of light polarization. Left panels: *y*-polarization channel. Right panels: *x*-polarization channel. The intensity is normalized to the maximum signal between the two channels. **g**, **l** Polarization anisotropy evolutions, over the first 1 ns, enclosed by the dashed boxes in (**f**, **k**). Only 6 representative snapshots are shown for clarity. **m** Normalized total fluorescence intensities and spatially averaged anisotropy for the 4 K molecule (top plots) and 20 K molecule (bottom plots) in the dual-cuvette configuration, calculated from (**k**–**l**). **h**–**j** Anisotropy lifetime maps of the imaged samples in the single-cuvette configuration. **n**–**o** Anisotropy lifetime maps of the imaged samples in the dual-cuvette configuration. Scale bars: 2 mm. The circles and arrows in (**c**–**m**) group the plots for either the left- or the right-*y* axis.

polarized in the *y*-axis, slices through the center of the cuvette. A BPF centered at 520 nm (transmission spectrum in Fig. S4) transmits around the emission peak at 515 nm (fluorescence spectrum in Fig. S10b) while eliminating residual excitation photons.

Figure 2c–e show the time-resolved plots of spatially integrated fluorescence signals in orthogonal polarization states (left *y*-axis) and spatially averaged anisotropy (right *y*-axis) of the three studied molecules. Time 0 on the *x*-axis is the time of fluorescence excitation. The light intensities are normalized to the maximum value of both polarizations. The PA (green lines) is derived from Eq. (12) in Methods. The black dashed lines are the exponential fits to the PA curves by

following Eqs. (13) and (14). A single exponential component is sufficient to describe the PA decay of small molecules of simple structures (i.e. FL and 4 K, plotted in Fig. 2c, d), while two exponential components are needed to fit to the PA dynamics of large molecules (i.e. 20 K, plotted in Fig. 2e). As can be seen, PAs of FITC-dextran molecules decay at lower rates than that of pure FL molecules. The Methods section contains detailed information on curve fitting and lifetime extraction. The inset photos in Fig. 2c–e are used to showcase the differences in fluorescence light levels between 4 K (inset of Fig. 2d) and FL and 20 K molecules in the insets of Fig. 2c, e, respectively. Under the same exposure condition, the 4 K photo shows greenish

image of fluorescence while the FL and 20 K photos show whiteish image due to saturation from very high intensity.

CUP2AI can capture 300 frames of single-pulse excited 2D anisotropy dynamics at 125 Gfps in a single shot. Figure 2f reveals the temporal image sequences of fluorescence intensity of both polarizations on an example of pure FL. Note that only 12 frames are selected for representation, while the full sequences can be found in Supplementary Movie 1. In Fig. 2g, we further plot the ultrafast spatiotemporal evolution of anisotropy. The PA decay time values at all pixels are then extracted and shown in Fig. 2h (for FL), together with the other two samples: 4 K and 20 K in Fig. 2i, j, respectively, following the same evaluation routine. Similarly, Fig. 2k–m show the 2D sequences and the corresponding 1D plots for a single shot measurement by simultaneously exciting two samples (4 K and 20 K) in two cuvettes. Figure 2n and o show the 2D PA lifetime maps for 4K–20 K and FL-20 K sample pairs, respectively. Note that 20 K molecule's anisotropy decay is almost 2 and 1.5 times slower than those of FL and 4 K molecules, respectively. It is also clear that despite some edge effects on the printed mask's fringes, the PA lifetimes are relatively uniform within the detection volume of one cuvette, indicating that molecules of the same kind behave consistently. Note that the lifetime maps do not resolve the letters well due to imperfect printing of the masks, allowing a small but detectable amount of light transmitting through the spacings between the letters. This is more obvious for FL and 20 K molecules whose fluorescence emissions are stronger. However, for 4 K molecules, the leaked light signals are too weak to estimate the anisotropy accurately. Therefore, the 4 K molecular lifetime map (left side of Fig. 2n) resembles the printed mask more than the others.

The rotational motion of molecules under ultrashort polarized irradiation is a direct reflection of the molecule's footprint and its surrounding environment (i.e. viscosity and temperature). When the environment condition is known, we can deduce the molecule's size from its PA lifetime [see Eq. (16) in Methods]. Figure 3a, b showcase CUP2AI's capability of 2D mapping of FL, 4 K, and 20 K fluorescent molecules in both single- and dual-cuvette configurations. Molecular volume images (Fig. 3a) are extracted from the anisotropy decay time maps (Fig. 2) using water's viscosity $\eta = 1.234 \times 10^{-3}$ Pa s and room temperature $T = 285$ K. Volume is further converted into molecular size (Fig. 3b) by assuming a simple spherical shape [see Eq. (18) in Methods]. The data points in Fig. 3c, d summarize the spatially averaged molecule volumes and molecule sizes, which are clustered with respect to the 3 different types of molecules. The volume values range from ~700 Å$^3$ for FL, over 1000 Å$^3$ for 4 K to 5500 Å$^3$ for 20 K molecules. The corresponding average sizes in terms of hydrodynamic diameter of FL, 4 K, and 20 K molecules are around 11.0 Å, 12.4 Å, and 21.8 Å, respectively. Both single- and dual-cuvette measurements show matching results. It is evident from the images in Fig. 3a, b that molecule sizes distribute homogeneously in the probed field-of-view. This observation also proves valid for the dual-cuvette setting where two types of molecules exhibit distinct but uniformly distributed size maps (right two columns in Fig. 3a, b). The spatial variations are quantitatively evaluated by plotting the error bars (SD, standard deviation) in Fig. 3c, d, in which the average SDs of FL's molecule volume and size are 200 Å$^3$ and 0.6 Å, respectively. The results from 4 K molecules have greater spatial non-uniformity owing to the worse signal-to-noise ratio (SNR) from the sample's lower fluorescence intensity caused by its lower quantum efficiency (see the inset photos in Fig. 2c–e, taken using the same exposure condition).

For each of the 5 measurement groups (single-cuvette FL, 4 K, 20 K, and dual-cuvette 4K-20K, FL-20K), we took 3 independent single-shot acquisitions, leading to the summary plots in Fig. 3c, d. Their reconstructed spatiotemporal intensity and PA dynamics are all shown in Supplementary Movies 1–5 with a frame interval as short as 8 ps. Besides the multiple 3D, 2D, and 1D plots presented in Figs. 2 and 3, the additional plots describing the remaining results (spatio-temporal

dynamics of intensity and anisotropy, 2D maps of anisotropy and molecule size) are given in Figs. S11–S15 in Supplementary Information.

As can be concluded qualitatively from the results in Figs. 2 and 3, fluorescent molecules of larger sizes (i.e. 20 K molecule is >4 K, 4 K molecule is larger than FL) tend to exhibit longer anisotropy lifetimes. One would expect slower Brownian rotation of molecules as the molecules grow larger and heavier. Therefore, longer times are required for more massive molecules to reorient their dipole moments and thus depolarize from the initial polarization aligned with the ultrashort excitation photon (along $y$-axis). This depolarization process due to Brownian rotation is illustrated in Fig. 3e. Our results are confirmed by the previous time resolved PA measurements performed using similar chemicals[11,12,31]. Further, the volume values of 4 K and 20 K molecules show an excellent agreement to the literature and theoretical calculations. For pure FL, the measured volume is larger than the expected value of 410 Å$^3$ based upon the molecular structure. Such a discrepancy can result from photo-dissociation and fluorescence saturation from the ultra-intense excitation pulse[33,34] and subsequent agglomeration[35], forming larger objects. When the molecule is too large, it no longer possesses regular shape so that its anisotropy evolution deviates from a simple mono-exponential decay[31]. For large molecules (e.g. 20 K), additional depolarization pathways caused by internal molecular motions start to take over, contributing to a shorter anisotropy decay than that caused by Brownian rotation. In this case, bi-exponential model is employed where the long decay constant is taken as the anisotropy lifetime used in sizing. See Methods and Eq. (14) for more descriptions. In Fig. 3f, the mono-exponential fit of 20 K molecule clearly shows a worse goodness-of-fit ($R^2$) than the other two types of molecules, supporting its anisotropy's departure from mono-exponential decay, while the bi-exponential fit exhibit a significant improvement in $R^2$ (from 0.78 to 0.97). Overall, CUP2AI allows correct mapping of diversely sized molecules in the same order of magnitude as the reference values and therefore can be used to discriminate and spatially locate various substances.

## PAH sizing using 50- and 25-Gfps PA imaging of 1PF in flames as a gaseous environment

Time-resolved PA signals have been used to investigate the formation of carbonaceous particles in flames[13], however these measurements are averaged and 0D (i.e. single point), therefore, limited in spatially resolving the laser-particle interactions in real-time. Moreover, in practical combustion devices, these particles or molecules are formed below nanometer range, requiring a much faster anisotropy approach and since the flames in gas phase are typically turbulent, single-shot methods are preferred. Conventional techniques that rely on scanning and sequential acquisitions over multiple pulses to capture spatiotemporal dynamics may fall short in accuracy due to the non-repeatability nature of turbulence. It is to be noted that using other established diagnostic techniques, such as laser-induced incandescence (LII)[36], it is hard to map nanoparticles smaller than ~5 nm, and using laser-induced fluorescence (LIF) it is impractical to quantify the sizes of PAH molecules which serve as precursors of incipient soot particles[37]. Therefore, CUP2AI bridges the gap between the gaseous PAH molecules and the solid soot particles, enabling investigations of the formation and growth of first incipient particles. Again, the depolarization of the detected signal depends on the size of the fluorescing molecule with larger molecules showing a slower anisotropy decay compared to smaller ones. One challenge compared to the measurements in liquid solvents, is that the Brownian motion in gases is typically much faster owing to low viscosity and therefore the decay times are overall shorter. This ultimately requires a higher imaging speed which comes at the cost of reduced SNRs. The benefit of using the streak camera in CUP2AI comes to rescue since it can readily work at a wide range of imaging speeds to balance temporal resolution and SNR. Specifically in this work, we also adopted $4 \times 4$-pixel binning in

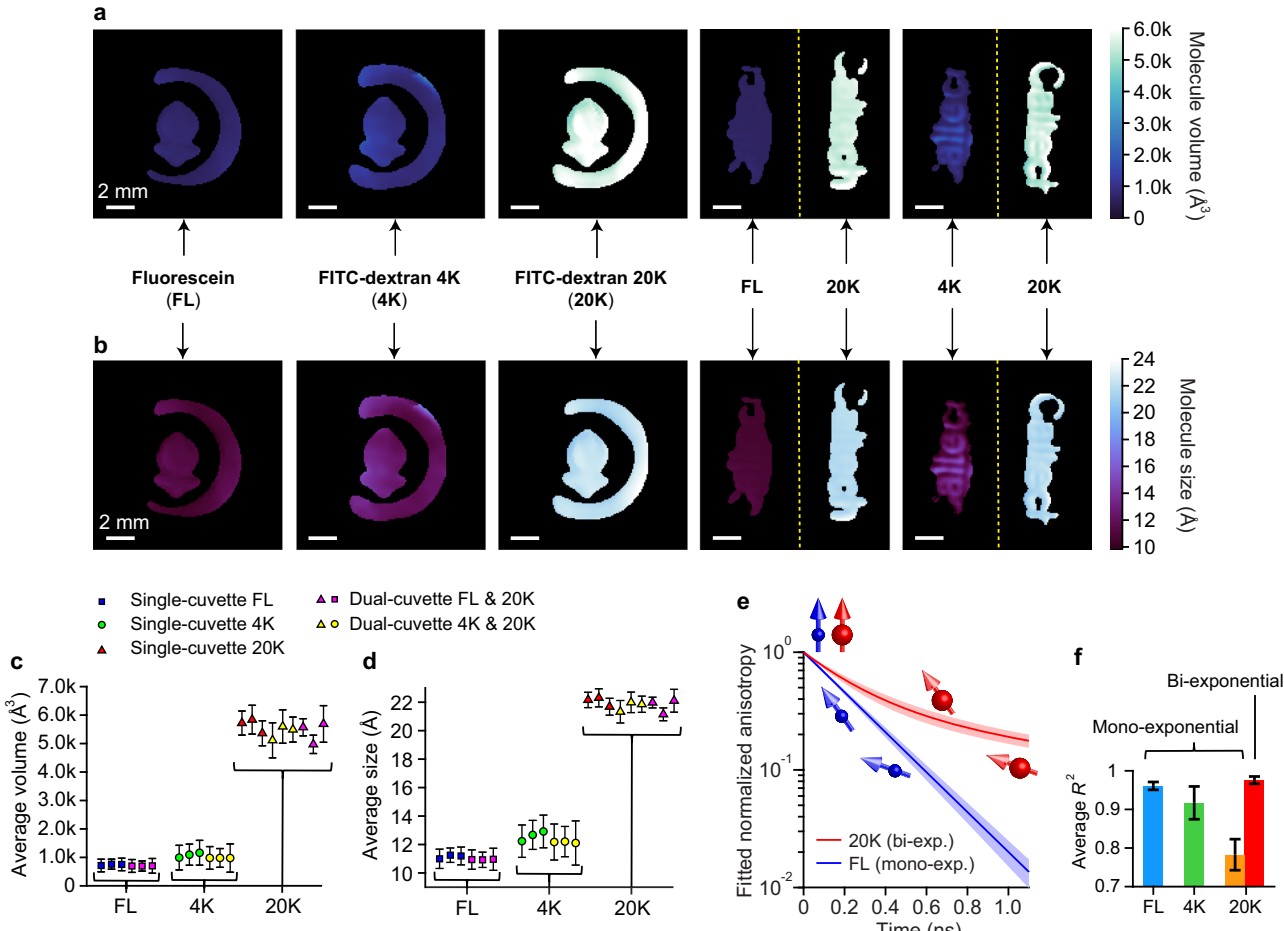

**Fig. 3 | Molecule size maps of fluorescent molecules in liquid form based on anisotropy lifetime maps from the 125-Gfps imaging results. a** Volumes of molecules. **b** Sizes of molecules. **a**, **b** From left to right: fluorescein, 4 K molecule, 20 K molecule, combination of fluorescein and 20 K molecule, combination of 4 K and 20 K molecules. Scale bars in (**a**) and (**b**): 2 mm. (**c**), Spatially averaged volumes of three types of molecules. (**d**), Spatially averaged sizes of three types of molecules. Error bars in (**c**, **d**) represent their standard deviations over space. **e** Illustration of how molecule size affects Brownian rotation and ultimately anisotropy decay rate. Exponential fit curves of polarization anisotropy decays of fluorescein (blue line) and 20 K (red line) molecules. They are averaged over all acquisitions and normalized to the maximum. The shaded patches represent standard deviations. Molecules are assumed to take spherical shapes, and the arrows point to the directions of dipole moments. The arrow directions are qualitatively drawn for illustration only. Note that mono-exponential fit (mono-exp.) is sufficient for small molecules with simple structures, while bi-exponential fit (bi-exp.) has to be applied for large molecules. **f** Goodness-of-fit of both mono-exponential and bi-exponential models, evaluated via the coefficient of determination ($R^2$), for the three molecules studied. These are averaged over all acquisitions, and error bars represent standard deviations. Blue and green bars are for fluorescein and 4 K molecules with mono-exponential fits, respectively, and orange and red bars are for the 20 K molecules with mono- and bi-exponential fits.

streak image to boost SNR. Additionally, a complex environment in the form of a mix of different PAHs and PAH clusters as intrinsic fluorophores in the measurement volume complicates the evaluation. However, to prove that the technique can be used in such a challenging environment, a simple non-premixed kerosene wick flame is investigated[30]. A schematic of the burner flame with the 400-nm UV laser sheet intercepting through the center is shown in Fig. 4a. A BPF with the center wavelength of 460 nm (transmission plot in Fig. S4) allows the LIF signals from PAH molecules to pass through in the imaging path. A relatively low fluence is used to avoid excitation of LII from soot nanoparticles.

A photograph of the flame is shown in Fig. 4b with a zoomed region revealing the spatial location and distribution of the LIF signal (blue colors) against the incandescence signal (greyscale) aside. As can be seen, the incandescence signal from solid soot particles surrounds the LIF signal deriving from PAHs as soot precursors. The growth process of these nanoparticles throughout the transition from gaseous to solid phase mainly follows the vertical main flow direction, however, due to diffusion also obeys a horizontal component. The

measurements of CUP2AI are therefore confined to the central region of the flame. In addition, the field-of-view (green dashed box in Fig. 4b) in the vertical direction covers 2 mm to 8 mm in height above the burner outlet (HAB), where soot inception from PAHs typically occurs. The imaging speed of the CUP2AI system was adjusted to achieve sufficient temporal resolution to resolve anisotropy lifetime of PAHs and meanwhile accommodate relatively lower intensity levels ascribed to PAH's low quantum efficiency. The recorded 25-Gfps sequences of orthogonal polarizations are selectively plotted in Fig. 4c, together with the PA sequence in Fig. 4d. The anisotropy climbs to the peak of ~0.4 at around time 0, as suggested in Fig. 4d, and then rapidly depolarizes to near 0 after a couple hundreds of picoseconds. The anisotropy lifetime map in Fig. 4e (including a zoom-in version in Fig. 4f), along with two vertical and two horizontal slices in Fig. 4g, h, reveal continuous rise in anisotropy lifetimes with increasing radial position as well as increasing HAB. The complete reconstructed spatiotemporal dynamics using both 25-Gfps and 50-Gfps frame rates are included in Supplementary Movies 6 and 7, respectively. Note that we acquired two independent CUP2AI datasets at each imaging speed.

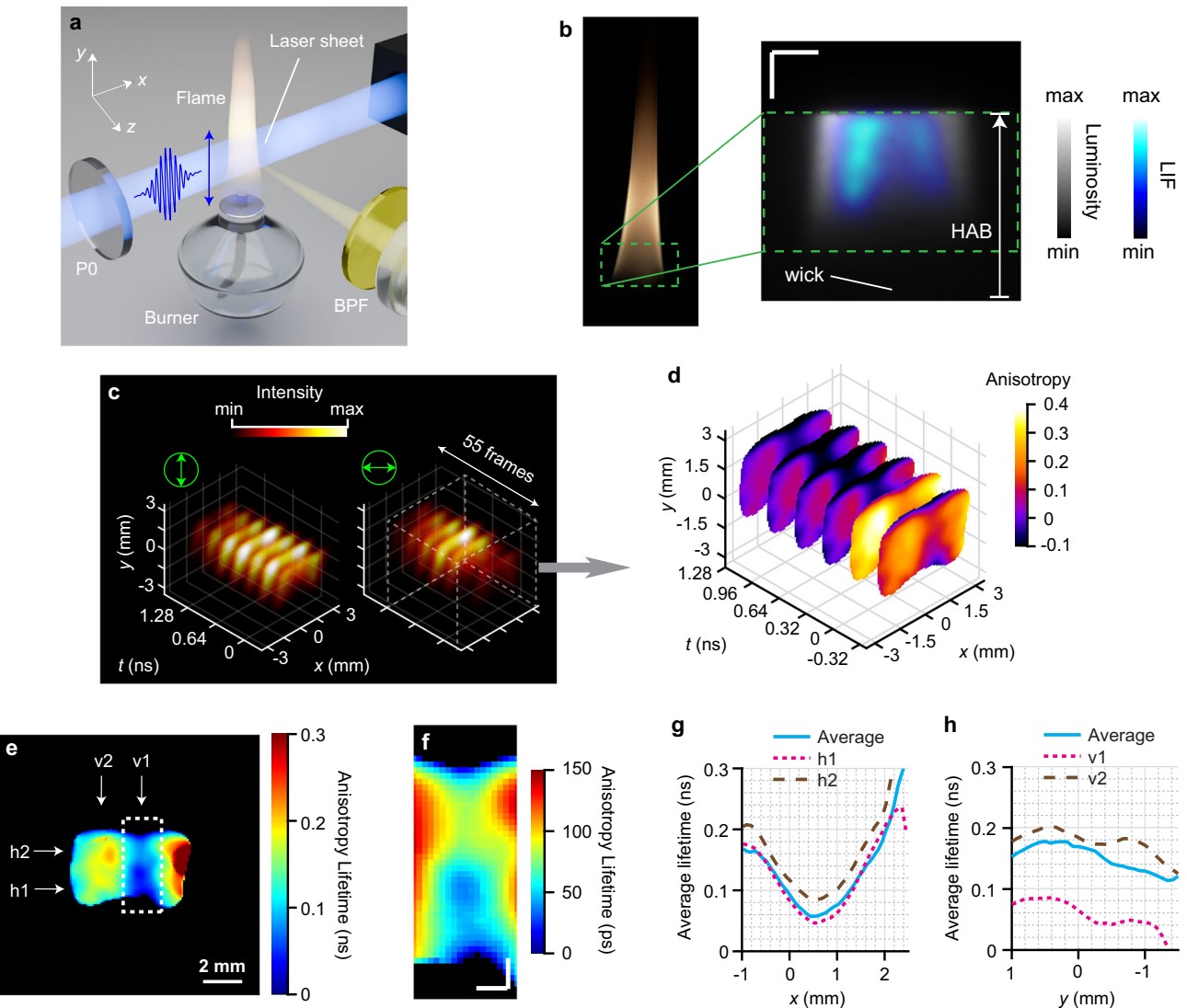

**Fig. 4 | ingle-shot imaging of laser-induced fluorescence (LIF) and polarization anisotropy from polycyclic aromatic hydrocarbon (PAH) molecules in a kerosene flame, excited by a single UV femtosecond laser pulse. S a** Schematic of the flame on the burner with the laser sheet passing through its center at the bottom. P0: linear polarizer, BPF: band-pass filter. The blue and yellow beams represent the fluorescence excitation and emission, respectively. **b** Left: a photograph of the flame studied. Right: a static camera image of the LIF signal overlaid on top of an image of the flame luminosity at the height above the burner (HAB), about 2 mm - 8 mm. The wick's position is labeled. Scale bars: 2 mm. **c–h** Results of 25-Gfps imaging. (**c**), Reconstructed intensity evolutions over 2.2 ns. In (**c**), 7 exemplary snapshots are selected among 55 frames. The green double-headed arrows represent the orientations of light polarization. Left panels: *x*-polarization channel. Right panels: *x*-polarization channel. The intensity is normalized to the maximum signal between the two channels. **d** Polarization anisotropy evolution, over the first 1.6 ns, enclosed by the dashed boxes in (**c**). Only 6 representative snapshots are shown for clarity. **e** Anisotropy lifetime maps of LIF-PAH. **f** Magnified views of LIF anisotropy lifetimes in the center regions of the flame, enclosed by dashed boxes in (**e**). The colormaps are rescaled. Scale bars in (**f**): 1 mm. **g** Average anisotropy lifetime along the *x*-direction (cyan solid line) and at two selected positions [h1 and h2 labeled in (**e**), in magenta short-dashed line and brown long-dashed line, respectively]. **h** Average anisotropy lifetime along the *y*-direction (cyan solid line) and at two selected positions [v1 and v2 labeled in (**e**), in magenta short-dashed line and brown long-dashed line, respectively].

Operations similar to those in the previous experiments (Fig. 3) are implemented to estimate the PAHs' molecule sizes. Here, when applying Eq. (15), we consider the flame to be as hot as $T = 900$ K and its dynamic viscosity $\eta = 4 \times 10^{-5}$ Pa s at this temperature. Results of two acquisitions at two imaging speeds are shown and analyzed in Fig. 5. Additional plots can be found in Figs. S16 and S17 in Supplementary Information. The absolute sizes ranging from 15 Å to 50 Å are in excellent agreement to previously reported values of ~3 nm[13] as well as ex situ investigations on similar precursor molecules[37]. To understand the impact of the imaging speed, a comparison between 50-Gfps (Fig. 5a–h) and 25-Gfps (Fig. 5i–p) is drawn and exhibits no significant difference in the general outcomes. However, we can still identify differences in the distributions of molecule sizes, especially in the

central region (Fig. 5b versus Fig. 5j, f versus Fig. 5n), and the differences are also obvious in the line plots (Fig. 5c, d versus Fig. 5k, l, and g, h versus Fig. 5o, p). Such shot-to-shot variations arising from slight flame movements manifest the unambiguous advantage of single-shot sensing. Thus, CUP2AI is proved to be a powerful and unique diagnostic tool that can non-invasively map differently sized molecules and particles with just a single laser pulse, which enables measurements even under chaotic conditions.

The 2D maps in Fig. 5a, e, i, and m exhibit increases in molecule volumes and sizes along both radial and axial directions. The directions of PAH growth are highlighted using orange arrows in Fig. 5c, d. The axial component of growth is further verified by the line plots in the *y*-direction (both averaged and at two positions v1 and v2 as labeled in

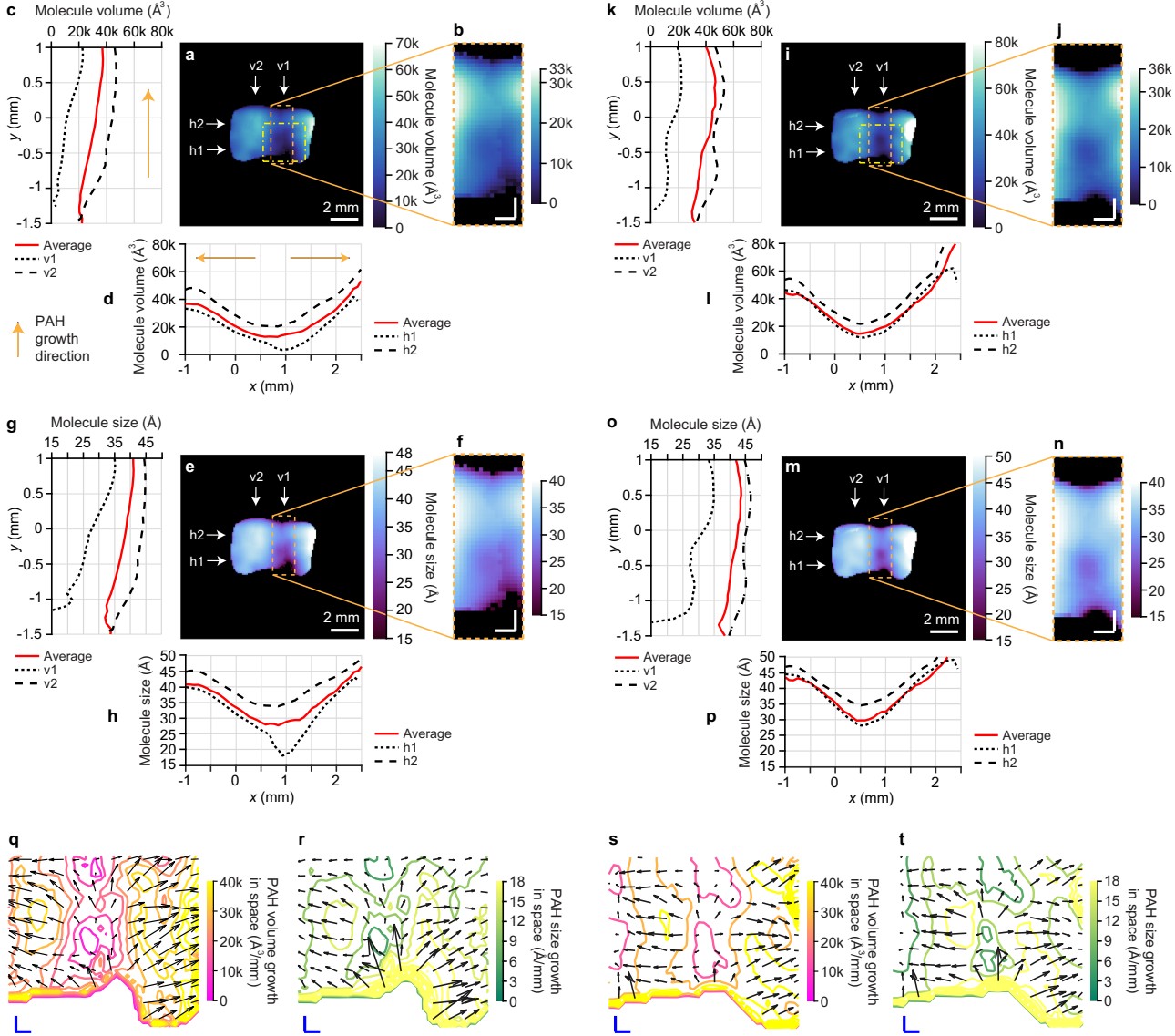

**Fig. 5 | Molecule size maps of PAH molecules in gaseous form based on LIF anisotropy lifetime maps from the 50-Gfps and 25-Gfps imaging results.** **a–h** Results of 50-Gfps imaging. (**i–p**), Results of 25-Gfps imaging. **a, i** 2D maps of the volumes of PAH molecules. **e, m** 2D maps of the sizes of PAH molecules. **b, f, j,** and **n** Magnified views of molecule volume and size maps in the center regions of the flame, enclosed by dashed boxes in (**a**), (**e**), (**i**), and (**m**). The color-maps are rescaled. Scale bars in (**a**), (**e**), (**i**), and (**m**): 2 mm. Scale bars in (**b**), (**f**), (**j**), and (**n**): 0.5 mm. **c, g** Average PAH volumes along the *y*-direction and at two selected positions [v1 and v2 labeled in (**a**, **e**)]. **d, h** Average PAH volumes along the *x*-direction and at two selected positions [h1 and h2 labeled in (**a**, **e**)]. **k, o** Average

PAH sizes along the *y*-direction and at two selected positions [v1 and v2 labeled in (**i**, **m**)]. **l, p** Average PAH sizes along the *x*-direction and at two selected positions [h1 and h2 labeled in (**i**) and (**m**)]. In (**c**, **d**) the directions of PAH molecule growth are marked by orange arrows. In (**c–p**) the red solid lines are for the spatially averaged values, and the black short-dashed and long-dashed lines are for two selected positions. **q, s** Contour plots of molecule volume growth in space by taking spatial gradients of (**a**) and (**i**). **r, t** Contour plots of molecule size growth in space by taking spatial gradients of (**e**) and (**m**). The lengths and directions of the black arrows in (**q–t**) represent the amplitudes and directions of spatial growth. Scale bars in (**q–t**): 0.25 mm.

Fig. 5a and e. Take the 50-Gfps case for example, as the *y* location moves upward from -1 mm to 1 mm (HAB increases), the PAH volume (Fig. 5c) expands from 4000 Å³ to 23,000 Å³ at the flame's center (v1), and from 30,000 Å³–47,000 Å³ near the flame's edge (v2). The corresponding PAH cluster size (Fig. 5g) grows from 18 Å to 35 Å and from 38 Å to 45 Å at v1 and v2, respectively. Similar analysis on radial growth shows that at the bottom of the flame (h1 in Fig. 5a and e), as the molecules diffuse outward by ~1.5 mm from the center (*x* = 1 mm), they enlarge in volume from about 3000 Å³ to 33,000 Å³ (Fig. 5d), equivalent to size growth from 18 Å to 40 Å (Fig. 5h). According to combustion theory, in the hot gas soup of molecules, PAHs harness more atoms (e.g. C, H) to form and add more benzene rings (length of one C-C bond is 1.4 Å in benzene) as they are pushed up from the burner and meanwhile move sideways via diffusion.

Figure 5q–t illustrate the lengths and directions of the black arrows, representing the amplitudes and orientations of spatial growth. These 2D maps depict the spatial evolution of PAH growth, ranging from 1.5 to 40 kÅ³ per mm in Fig. 5q and s, and from 1 to 18 Å³ per mm in Fig. 5r and t, respectively. The 2D molecule sizing by CUP2AI hence confirms the model of molecular growth of soot precursor structures in a non-premixed diffusion flame.

**Molecule sizing using 12.5-Gfps PA imaging of 2 PF in a liquid environment**

In comparison to 1PF anisotropy measurements, the 2P excited anisotropy has been reported to provide enhanced sensitivity[5] and other features such as distinct measurement volume as the process

only occurs at locations of high photon densities, thus background or out-of-focus light is suppressed. Additionally, in many applications such as medical diagnosis or treatment, it is impractical to use UV excitation since it might damage the probe and the environment, and it has shallow penetration depths caused by strong scattering. Here, we implemented CUP2AI for 2PF anisotropy in liquid solutions using 800-nm NIR excitation at 12.5-Gfps imaging speed in a single shot using both single- and dual-cuvette arrangements, as illustrated in Fig. 6a, b, respectively. Due to lower emission intensity in 2PF than in 1PF, the imaging speed is compromised to gain sufficient SNR. Figure 6o shows the averaged anisotropy lifetimes for single and dual cuvette measurements of the same FL, 4 K, and 20 K samples as in 1 P excitation. Figure 6c–e, f–h, j–k show spatiotemporal intensity and anisotropy dynamics and anisotropy lifetime maps in single-cuvette measurements for FL, 4 K, and 20 K molecules, respectively. Similarly, Fig. 6l–n are the results of dual-cuvette measurements (FL and 20 K). The entire dynamics of the 2P-induced processes (totally 100 frames in single acquisition) can also be followed in Supplementary Movie 8. Plots in Fig. 6c, f, i, and l show varying rates of anisotropy decay, with the FL molecule being the fastest and the 20 K molecule being the slowest. This is again confirmed by the time-resolved stacks of selected anisotropy images in Fig. 6d, g, j, and m. A slightly higher anisotropy from FL with 2 P excitation in comparison to 1 P is consistent with measurements reported in refs. 5,11. Similar to the results of 1PF in Fig. 2, the PA lifetime maps in Fig. 6e, h, k, and n demonstrate homogeneous spatial distributions and lengthened decay times for molecules of augmented footprints. Spatially averaged lifetimes of FL and 4 K molecules, plotted in Fig. 6o, are longer than those obtained from 1PF experiments primarily due to decreased temporal resolution with lower imaging speed and poorer SNR. However, the data is in the same order of magnitude and exhibits the same general trend. Finally, both molecule volumes and sizes are calculated and then analyzed, as plotted in Fig. S19 in Supplementary Information. The spatially averaged size of 20 K molecule (21.5 Å) is within a 1.5% difference from that obtained using 1PF (see Fig. 3d). The larger spatial variations in 2PF compared to those in 1PF are primarily due to lower SNR in raw data and worse accuracy in numerical deconvolution resulting from a broader temporal response (see Fig. S8 in the Supplementary Information). The overall agreement between the results from 2PF and 1PF validates the usefulness of CUP2AI for 2PF anisotropy, making it a viable option in cases where 1PF cannot be applied (e.g. less material damage, deeper detection under surface, and higher spatial resolution are desired).

## Potential applications of CUP2AI

In the previous sections, we successfully demonstrated single-shot 2D molecule sizing for the cases of liquid and gaseous solvent environments, one and two photon excitations and extrinsic as well as intrinsic fluorophores. While the work presents the CUP2AI system effectively deployed in three exemplary scenarios, the potential use of this technique is by far not limited to these conditions. In a next logical step, transient fluid processes with an underlying molecular size distribution can be investigated in situ on a single shot basis, offering renewed understanding, which are currently not accessible. These include, e.g., spray drying processes in pharmaceutical applications[38] or the gas phase synthesis of nanoparticles or quantum dots[39]. CUP2AI can be integrated with structured laser sheet-based FRAME approach for ultrafast and high-contrast visualization[40]. Additionally, chemical transitions, growth, and decomposition reactions, which often take place in liquid solvents, can be rapidly and quantitatively analyzed. The example of the flame environment can be readily extended to more technical and turbulent flame conditions, providing valuable insights into growth and inception processes, which are currently not fully understood. Here,

another possible application relates to the fluorescence lifetime itself as it correlates various flame sampled materials with their molecular weights[29]. Additionally, CUP2AI's single-shot capability appears as a distinct advantage since among three basic phases of matter, liquid and gas experience more violent disturbance thanks to weak friction forces. Finally, in biomedical imaging, CUP2AI can perform rapid non-invasive and real-time imaging using two-photon scanned light-sheet microscopy[15]. Thus, real-time monitoring of transient phenomena in fluid circumstances becomes crucial. This renders CUP2AI enormous prospects in molecule research in the fields of biological, health, and pharmaceutical studies. To sum up, CUP2AI is a powerful and one-of-its-kind real-time ultrafast imaging and molecule sizing tool that holds great potential to enable next-phase scientific discoveries and technological innovations.

## Methods

### Streak camera

In a streak camera[41], as schematically illustrated in Fig. S5(a), photons carrying spatiotemporal information are converted to electrons by a photocathode. The times the electrons leaving the photocathode ($t_1$, $t_2$, $t_3$ in Fig. S5a) follow the times of arrival of the original photons. After acceleration by a high-voltage anode, the photoelectrons are redirected in the $y_s$ direction (vertical) by a fast-varying voltage (inset of Fig. S5a). Therefore, when collected by a phosphor screen, the electrons coming from different times [$e(t_1)$, $e(t_2)$, $e(t_3)$] are located at different vertical positions. The displacement between adjacent electrons [e.g. $e(t_1)$ and $e(t_2)$] are determined by the rate of the sweeping voltage. An image intensifier amplifies the spatio-temporally integrated image on the phosphor screen, which is eventually recorded by an internal CMOS sensor.

To synchronize fluorescence excitation and streak camera's acquisition, a trigger signal (500 Hz) from the femtosecond laser is input to a delay generator (Stanford Research, DG645), which outputs an under-sampled (10 times) signal with an optimized delay to trigger the streak camera. To enhance SNR in raw images, we employed 4 × 4-pixel binning.

### Digital micro-mirror device

The DMD is a powerful instrument to spatially modulate light beams and finds diverse applications in modern optics[42,43]. The DMD (Texas Instruments, LightCrafter 3000) used in this work is made of aluminum-coated square micro-mirrors, arranged diagonally in an array of 608 columns and 684 rows. Each mirror is independently addressed by the corresponding single pixel in the image input to the device. In the simple scenario of displaying binary patterns, the individual micro-mirror can be electronically actuated to turn to either +12° (1 in binary code) or −12° (0 in binary code), reflecting two complementarily patterned beams in the directions +24° or −24° away from the original incident beam. Here, 4 × 4 DMD pixels are grouped to form one spatial encoding pixel (i.e. one pixel in Fig. 1c and Fig. S1b). In compressed-sensing-based imaging, this encoding size (i.e. binning of DMD elements) is proportional to the spatial and temporal resolutions[44]. Therefore, a smaller binning factor is desired for a higher resolution. However, the SNR and space-charge effect of the streak camera compromise the contrast of the images when a small DMD binning is used. As a result, we need to trade off our choice to a sweet spot of 4 × 4 binning. A similar study on another streak camera can be found in the previous work[27]. Note that we use a pattern with 50% 1-code and 50% 0-code to balance the reflected light intensities in the two complementary encoding channels.

### Liquid fluorescence sample preparation

The molecules came in the form of powders (Sigma-Aldrich) and were properly weighted on an electronic balance. Then they were separately

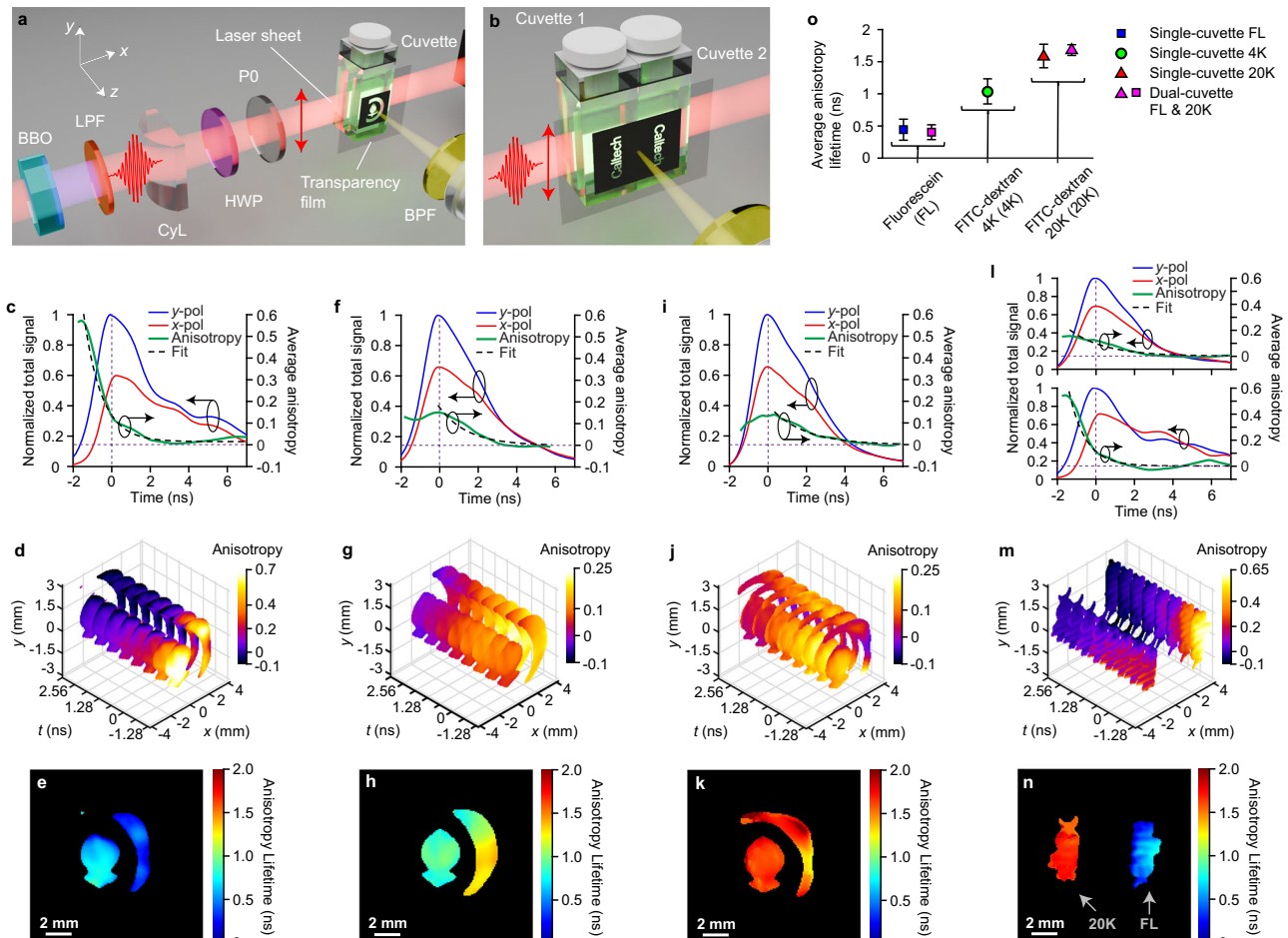

**Fig. 6 | 12.5-Gfps polarization anisotropy imaging of fluorescein, 4 K, 20 K molecules and their combinations, excited by a single NIR femtosecond laser pulse via the two-photon process. a** Schematic of a single cuvette with one type of fluorescence molecule and a single pattern. BBO: beta barium borate, BPF: band-pass filter, CyL: cylindrical lens, HWP: half-wave plate, P0: linear polarizer, LPF: long-pass filter. **b** Schematic of dual cuvettes side-by-side with two types of fluorescence molecules and two patterns. In (**a**) and (**b**), the red and yellow beams represent the fluorescence excitation and emission, respectively. A long-pass filter is employed to select the 800 nm light for two-photon fluorescence, replacing the short-pass filter used in one-photon fluorescence in Fig. 2a, b. Figure 6o shows the averaged anisotropy lifetimes for single and dual cuvette measurements of the same FL, 4 K, and 20 K samples as in 1 P excitation. In (**c–n**), results of (**c–e**) fluorescein molecule,

(**f–h**) 4 K molecule, (**i–k**) 20 K molecule in the single-cuvette configuration, and (**l–n**) fluorescein and 20 K molecules in the dual-cuvette configuration. In (**c**), (**f**), (**i**), and (**l**), Left *y*-axis: evolutions of normalized spatially integrated intensities from both *x*- and *y*-polarization channels (red and blue solid lines). Right *y*-axis: evolutions of spatially averaged anisotropy. The green solid lines are the measured anisotropies, and the black dashed lines represent exponential fits. The circles and arrows group the plots for either the left- or the right-*y* axis. In (**l**), top plots are for 20 K molecule and bottom plots are for fluorescein. **d**, **g**, **j**, and **m** Polarization anisotropy evolutions, over the first 4.5 ns. Only 8 representative snapshots are shown for clarity. **e**, **h**, **k** and **n** Anisotropy lifetime maps of the imaged samples. **o** Spatially averaged anisotropy lifetimes of three types of molecules. Error bars represent their standard deviations over space.

---

mixed with pure distilled water to make a concentration of 50 μM, facilitated by slight agitation. Their molecular masses, powder weights and webpages from the vendor are summarized in Table S4. About 1 mL of the solutions were then transferred to three surface-polished quartz cuvettes (Thorlabs, CV10Q14FA) by clean pipettes. The Caltech logo patterns were printed on sheets of transparency film by a black-white printer and applied on the front surfaces of the cuvettes using Scotch tape.

### Image formation model

The total fluorescence emission collected by the imaging system in the spatio-temporal domain can be decomposed into two orthogonal polarizations, as described by

$$I_0(x, y, t) = I^{\parallel}(x, y, t) + I^{\perp}(x, y, t) \quad (1)$$

In Eq. (1), $I^{\parallel}$ and $I^{\perp}$ stand for polarizations parallel (*y*-direction) and perpendicular (*x*-direction) to that of the excitation laser pulse,

respectively. Their representations are

$$I^{\parallel}(x, y, t) = \boldsymbol{P}^{\parallel}\boldsymbol{I} = \begin{bmatrix} 1 & 0 \end{bmatrix} \cdot \begin{bmatrix} I^{\parallel}(x, y, t) \\ I^{\perp}(x, y, t) \end{bmatrix}, \quad (2.1)$$

$$I^{\perp}(x, y, t) = \boldsymbol{P}^{\perp}\boldsymbol{I} = \begin{bmatrix} 0 & 1 \end{bmatrix} \cdot \begin{bmatrix} I^{\parallel}(x, y, t) \\ I^{\perp}(x, y, t) \end{bmatrix}, \quad (2.2)$$

where $\boldsymbol{I} = \begin{bmatrix} I^{\parallel}(x, y, t) \\ I^{\perp}(x, y, t) \end{bmatrix}$ contains the orthogonal polarization components of the emitted fluorescence light, $\boldsymbol{P}^{\parallel}$ and $\boldsymbol{P}^{\perp}$ are the operators of the two polarizers, P1 and P2, whose transmission axes are in the *y*-direction and the *x*-direction, respectively, in the two imaging paths before the streak camera (see Fig. 1d and Fig. S1c).

There are totally three images captured in one CUP2AI acquisi-tion: one time-unsheared image by the external CCD camera and two

time-sheared images by the streak camera, represented by $E_0$, $E_1$, and $E_2$, respectively. To express how CUP2AI images are formed, we assign operators $C_1$ and $C_2$ to represent complementary spatial encoding by DMD (see Fig. 1c and Fig. S1b). Inside the streak camera, temporal shearing by the sweeping voltage and spatio-temporal integration by the phosphor screen can be denoted by operators $S$ and $T$. A visualization of these two operators is illustrated in Fig. S5b above. After considering spatial low-pass filtering ($F_0$, $F_1$, and $F_2$ for the three views) and distortion ($D_1$ and $D_2$ for the two time-sheared views) operators due to optical components, we can fully depict CUP2AI's imaging model by

$$E_0 = TF_0 I^\parallel + TF_0 I^\perp \tag{3.1}$$

$$\alpha_1 E_1 = TSP_1 D_1 F_1 C_1 I^\parallel \tag{3.2}$$

$$\alpha_2 E_2 = TSP_2 D_2 F_2 C_2 I^\perp \tag{3.3}$$

Here, low-pass filtering, owing to the finite aperture and wavefront aberrations of the imaging optics, blurs the image, while geometric distortion, caused by aberration and slight misalignment, warps and rotates the image. Equations (3.1)–(3.3) are re-written in the matrix form

$$\begin{bmatrix} E_0 \\ \alpha_1 E_1 \\ \alpha_2 E_2 \end{bmatrix} = \begin{bmatrix} TF_0 & TF_0 \\ TSD_1 F_1 C_1 & 0 \\ 0 & TSD_2 F_2 C_2 \end{bmatrix} \cdot \begin{bmatrix} I^\parallel \\ I^\perp \end{bmatrix} \tag{4}$$

In Eq. (4), $\alpha_1$ and $\alpha_2$ are the weighting factors in the two time-sheared views relative to the time-unsheared view. They are used to balance transmission discrepancy in the optical components between different viewing channels. Note that none of the operators in Eq. (4) changes the polarization state of light. For simplicity, the forward model above can take a concatenated format,

$$E = OI, \tag{5}$$

in which $E$ contains all acquired images and $I$ contains spatio-temporal datasets to solve. It is crucial to accurately calibrate all the operators in $O$. To calibrate for $F_1$, $C_1$, $F_2$, and $C_2$, we took an image of the encoding masks when the imaging system was flood illuminated by a uniform beam (collimated green laser diode) and the sweeping voltage in the streak camera was turned off (operated as a conventional camera). Using the same experimental configuration, regular patterns (e.g. checkerboard or dot array) were then imaged to calibrate for the distortions $D_1$ and $D_2$. The temporal shearing operations ($S$) for different imaging speeds are provided by the streak camera manufacturer. The experimental procedures are also detailed in our previous efforts[23,28]. A simplified depiction of image formation can be found in Fig. S5b.

## Image reconstruction method

To recover $I^\parallel$ and $I^\perp$ from the captured images $E$ and the calibrated operators $O$, we solve an ill-posed inverse problem by resorting to regularization, equivalent to solving the following optimization problem,

$$\mathrm{argmin}_I \left\{ \frac{1}{2} \| E - OI \|_2^2 + \omega \Phi_{TV}(I) \right\} \tag{6}$$

In Eq. (6), the measurement fidelity (the first term) and the regularizer (the second term) are balanced by an optimized parameter $\omega$. The regularizer promotes sparsity in the solution and assists reconstruction to converge to global minimum[45–47]. Most real-world ultrafast

phenomena satisfy this assumption of sparsity, which has been experimentally verified[26].

In this work, total variation is employed as the regularizer, which contains two components: $\Phi_{TV}(I) = \Psi(I^\parallel) + \Psi(I^\perp)$, corresponding to intensities from two orthogonal polarizations for 3D data cube. Hence, $\Psi(I^\parallel)$ and $\Psi(I^\perp)$ are expressed by Eqs. (7.1) and (7.2) as:

$$\Psi(I^\parallel) = \sum_t \sum_{x,y} \sqrt{\left[ I^\parallel(x+1,y,t) - I^\parallel(x,y,t) \right]^2 + \left[ I^\parallel(x,y+1,t) - I^\parallel(x,y,t) \right]^2}$$
$$\tag{7.1}$$

$$\Psi(I^\perp) = \sum_t \sum_{x,y} \sqrt{\left[ I^\perp(x+1,y,t) - I^\perp(x,y,t) \right]^2 + \left[ I^\perp(x,y+1,t) - I^\perp(x,y,t) \right]^2}$$
$$\tag{7.2}$$

Here, we adopted the two-step iterative shrinkage/thresholding (TwIST) algorithm[48] iteratively solve the minimization problem in Eq. (6). The TwIST algorithm and the use of total variation as the regularizer have been extensively implemented in the previous Coded Aperture Snapshot Spectral Imaging (CASSI) technique[46,47,49]. In this current work, both raw images (captured by the streak camera and the external camera) and the calibrated operators in matrix $O$ are inputs to the TwIST algorithm. The regularizer function needs to be modified according to TV's definition in Eqs. (7.1) and (7.2). The regularization parameter $\omega$ and the weighting factors $\alpha_1$, $\alpha_2$ are optimized and selected based on a brute-force searching step in which wide parameter spaces are scanned. The $\omega$ parameter is typically around $10^{-4}$, while $\alpha_1$ and $\alpha_2$ fall between 0.1 and 0.5. Two stop criteria are employed: (1) the maximum number of iterations is 100; (2) the minimum improvement in the objective function [defined in Eq. (6)] after one iteration is set to be $10^{-3}$. The reconstruction algorithm terminates when either criterion is satisfied. In the future, we aim to explore various optimization parameters and matrix conditions within the compressed sensing framework.

## Fluorescence anisotropy and lifetime extraction

As laid out in detail by Pu et al.[31], the total fluorescence intensity $I$ as a function of time $t$ can be described by an exponential decay with the decay lifetime $\tau_f$

$$I(t) = I_0 \exp \left( -\frac{t}{\tau_f} \right) \tag{8}$$

and the initial fluorescence intensity is $I_0$ at $t = 0$. Further, $I(t)$ can be rephrased in terms of the polarization components parallel (denoted by the superscript $\parallel$) and perpendicular (superscript $\perp$) to the excitation following

$$I(t) = I^\parallel(t) + 2 I^\perp(t), \tag{9}$$

where the factor of 2 accounts for the two perpendicular components. Anisotropy is defined by the polarization ratio of expression as

$$a(t) = \frac{I^\parallel(t) - I^\perp(t)}{I(t)} = \frac{I^\parallel(t) - I^\perp(t)}{I^\parallel(t) + 2 I^\perp(t)}, \tag{10}$$

which itself follows an exponential decay function given by

$$a(t) = a_0 \exp(-\theta t). \tag{11}$$

Here, $a_0$ is the initial polarization anisotropy at $t = 0$ and $\theta$ is the anisotropy correlation exponent with its inverse revealing the anisotropy decay time $\tau_a = 1/\theta$.

After CUP2AI's reconstruction by solving Eq. (6), we can obtain two 3D data cubes in Eq. (1): $I^{\parallel}(x,y,t)$ and $I^{\perp}(x,y,t)$. Then, it becomes straightforward to compute the spatiotemporal dynamics of polarization anisotropy

$$a(x,y,t) = \frac{I^{\parallel}(x,y,t) - I^{\perp}(x,y,t)}{I^{\parallel}(x,y,t) + 2\,I^{\perp}(x,y,t)}. \quad (12)$$

Here, $a(x,y,t)$ is assumed to follow either a mono-exponential decay based on Eq. (11) or a bi-exponential decay

$$a(x,y,t) = a_0(x,y)\exp\left(-\frac{t}{\tau_0(x,y)}\right) \quad (13)$$

$$a(x,y,t) = a_1(x,y)\exp\left(-\frac{t}{\tau_1(x,y)}\right) + a_2(x,y)\exp\left(-\frac{t}{\tau_2(x,y)}\right) \quad (14)$$

Therefore, at each spatial point $(x,y)$, we can apply nonlinear least square fitting (use the lsqnonlin function in MATLAB) to extract $\tau_0(x,y)$ or $\tau_1(x,y)$ and $\tau_2(x,y)$ from $a(x,y,t)$.

The mono-exponential model in Eq. (13) can only accurately describe anisotropy dynamics of small molecules with simple structures, such as fluorescein and FITC-dextran 4 K molecules. More details of these exponential decays can be found in Chapter 10.1.4 in ref. 2. Therefore, these two molecules studied in this work have anisotropy lifetimes defined by $\tau_a(x,y) = \tau_0(x,y)$.

As molecules grow bigger, gaining more complicated structures, a mono-exponential curve no longer follows the anisotropy dynamics. In these large molecules, there exist at least two anisotropy decay processes: one with a shorter lifetime [i.e. $\tau_1$ in Eq. (13)] and the other with a much longer lifetime [i.e. $\tau_2$ in Eq. (14)]. Usually, the short decay term originates from internal movement of molecules, while the long decay corresponds to Brownian rotation, contributing to the molecule size estimation. For the FITC-dextran 20 K molecule, we need to apply the bi-exponential model instead (see Fig. 3f). Hence, the anisotropy lifetime of the 20 K molecule is defined as $\tau_a(x,y) = \tau_2(x,y)$. It is noteworthy that if bi-exponential fitting is applied to fluorescein or 4 K molecules, the initial anisotropy of the long-decay term ($a_2$) ends up being nearly zero.

Note that this anisotropy decay time (or lifetime) $\tau_a(x,y)$ results from the convolution of the actual anisotropy decay with the CUP2AI's system temporal response. Please see Section 5.1 in Supplementary Information for the calibrated temporal PSFs. Hence, a numerical deconvolution process is implemented to obtain the original anisotropy lifetime maps $\tau_a^{\text{orig}}(x,y)$ from the raw measurements $\tau_a(x,y)$ (refer to Figs. S6–S8 in Supplementary Information). To be specific, a look-up-table (i.e. a calculated plot) is used to infer deconvolved lifetime from the measured lifetime. The data cubes $I^{\parallel}(x,y,t)$, $I^{\perp}(x,y,t)$, $a(x,y,t)$, and time-corrected lifetime maps $\tau_a^{\text{orig}}(x,y)$ are shown in Fig. 2, Fig. 4, Fig. 6, Figs. S10–S17, and Movies S1–S8.

## Molecule size modeling

Once the anisotropy decay time $1/\theta$ is extracted from the measurements, it can be used to infer the molecular volume $V$ following the Stokes-Einstein-Debye relation[2].

$$V = \frac{1/\theta k_B T}{\eta} \quad (15)$$

Therefore, the spatial distribution of molecular volume can be expressed by

$$V(x,y) = \frac{\tau_a(x,y) k_B T}{\eta} \quad (16)$$

Here, $k_B = 1.38 \times 10^{-23}\,\text{m}^2\,\text{kg}\,\text{s}^{-2}\,\text{K}^{-1}$ is Boltzmann's constant, $\eta$ the dynamic viscosity and $T$ the temperature. For the investigated samples, the values of the latter two parameters take $\eta = 1.234 \times 10^{-3}\,\text{Pa s}$, $T = 285\,\text{K}$ for the deionized water solvent, and $\eta = 4 \times 10^{-5}\,\text{Pa s}$, $T = 900\,\text{K}$ for the flame case[13]. Note that we assume both temperature and dynamic viscosity are constant over the field-of-view in the samples. Especially for the flame case as a complex environment of multiple PAHs and flame formed particles these are strong assumptions impacting the results, which in this context should be interpreted in a semi-quantitative manner. Additionally, the molecular volume typically is expressed in terms of angstrom cubic and can be used to infer the mean molecular size assuming it to be mostly spherical.

The volume of a sphere is expressed by

$$V = \frac{4}{3}\pi r^3, \quad (17)$$

in which $r$ is the sphere's radius. Therefore, the hydrodynamic diameter (diameter $d = 2r$) distribution (image coordinates $x$, $y$) of the molecule can be calculated from its volume distribution by

$$d(x,y) = \sqrt[3]{\frac{6}{\pi}V(x,y)}, \quad (18)$$

## Reporting summary

Further information on research design is available in the Nature Portfolio Reporting Summary linked to this article.

## Data availability

The data that support the findings of this study are available from the corresponding author on request. An example dataset is included with the example code deposited in Code Ocean.

## Code availability

The reconstruction algorithm is described in detail in Supplementary Information. Example data and processing codes can be accessed via Code Ocean. The example codes and data convert spatiotemporal (3D) movies of fluorescence intensity to 2D maps of molecular volumes and diameters.

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

## Acknowledgements

P.W. was supported by National Institutes of Health grants R35 CA220436 and R01 EB028277. Y.N.M. gratefully acknowledges the Swedish Research Council for the financial support of grant # IPD2018-06783. Part of the work of Y.N.M. that was enabled by the Jet Propulsion Laboratory, under a contract with NASA (80NM0018D0004). M.S.G. thanks funding from JPL JROC and NASA DDAP and NFDAP programs. The authors thank Dr. Paresh Kumar Samantaray for providing the fluorescein powder.

## Author contributions

P.W. and Y.N.M. built and characterized the CUP2AI system. P.W. and Y.N.M. performed the experiments. P.W. and Y.N.M. conducted the image reconstruction. F.J.B. proposed the idea of imaging polarization anisotropy in flames and liquid media. Y.N.M proposed the idea of two photon fluorescence excitation. F.J.B. performed numerical modeling.

P.W. organized and managed the project. P.W., Y.N.M., and F.J.B. performed data analysis and drafted the manuscript. P.W. worked on data visualization. M.S.G. reviewed the technical details and advised on evaluation approaches. L.V.W. supervised the project. All authors revised the manuscript.

## Competing interests

The authors disclose the following patents for the CUP technology: WO2016085571 A3 (L.V.W.), US10992924B2 (L.V.W. and P.W.), and US11561134B2 (L.V.W. and P.W.). The remaining authors declare no competing interests.
