## [Transparent Peer Review file · Nature Communications]

Single-shot two-dimensional nano-size mapping of fluorescent molecules by ultrafast polarization anisotropy imaging

Corresponding Author: Professor Lihong Wang

Version 0:

Reviewer comments:

Reviewer #1

(Remarks to the Author)

Please see the attached file.

(Remarks on code availability)

Reviewer #2

(Remarks to the Author)

The author demonstrated the world's fastest fluorescence polarization anisotropy imaging using their streak camera. The analysis method is new and powerful, the physics and the data are profound and solid. I therefore support its publication in Nature Communications. I have only one comment:

The data acquisition is so fast so that for each molecule, there is even less than one photon detected in each frame. So, the result is still an ensemble average. What is the new information this ultrafast method can achieve when compared with those methods that capture the time-domain average signal (such as the method reported in ref. 12)? Since the average on the time domain is more or less equivalent to the average over many molecules (for an ergodic random process). The authors may need to explain this concern slightly.

(Remarks on code availability)

Reviewer #3

(Remarks to the Author)

This work provided a new method of polarization anisotropy, named compressed ultrafast planar polarization anisotropy imaging (CUP2AI). The in-situ, transient and 2D measurement of molecule size both in liquid and gas phase is achieved by leveraging rotational diffusivity of molecules, femtosecond laser-sheet illumination, and compressed sensing. The CUP2AI provides a good evaluation of sizes of fluorescein in water and polycyclic aromatic hydrocarbons in flames. This work would benefit the chemical, biological and even engineering community.

However, I have some comments for the authors:

(1) The measurement is based on the model of Stokes-Einstein-Debye relation with the parameters of anisotropy decay time, temperature and viscosity. The decay time is fitted by the CUP2AI measurement result. Could you please evaluate the influence of parameter errors?

(2) According to Fig. S2 and Fig. S3, the laser intensity is not uniform distribution among the 2D view. Does the laser intensity distribution cause the measurement error? Does the parameter of anisotropy decay time depend on the laser intensity?

- (3) The real molecules are not sphere. The results of CUP2AI provide the hydrodynamic diameter. What is the relationship between the real morphology and the diameter?
- (4) $a(x,y,t)$ is assumed to follow either a mono-exponential decay or a bi-exponential decay, according to Eq. (11)-(14). Why? In Fig. 2c and Fig. 2d, the anisotropy is fitted by mono-exponential decay, but the anisotropy is fitted by bi-exponential decay in Fig. 2e. What is the reason for fitting by different functions?
- (5) As for the PAH measurement in the flame, the non-premixed kerosene wick flame could be unstable. Do the authors try other stable flames? Do the authors compare the CUP2AI results and previous literature investigations?

(Remarks on code availability)

Reviewer #4

(Remarks to the Author)

Peer Review of "Single-shot two-dimensional nano-size mapping of fluorescent molecules by ultrafast polarization anisotropy imaging"

In this manuscript, the authors present a new imaging method extending the technique of compressed ultrafast photography to spatially mapping polarization anisotropy by wide-field imaging of fluorescence generated by a femtosecond excitation pulse. Under some assumptions about the rotational dynamics of the emission dipole, polarization anisotropy measurements are translated to measurements of molecular size via determination of the anisotropy decay time. The new technique is characterized using proof-of-principle experiments with fluorophore solutions and a more interesting demonstration with laser-induced fluorescence from hydrocarbons formed in a kerosene flame.

The results presented in this work are novel and have clear application to nanoscale size measurement in a variety of experimental contexts. Two-dimensional characterization of transient species in flames and aerosols as done in this paper is of particular interest since this is a challenging experimental regime. The use of a single optical excitation pulse is also of special interest for light-sensitive environments.

The manuscript is impressively thorough in detailing the experimental design, analysis methodology, and results. The detailed schematics and parts list in the Supplementary Materials promotes transparency and reproducibility, and the extensive review of different aspects of the experiment throughout the text make it easy to follow exactly what the authors did.

I have only a few minor remarks which I believe would improve the manuscript:

- I found the ovals with arrows in e.g. Figs. 2, 6, S4, and S11-17 confusing. Please add a statement about them to the figure caption, or consider an alternative means of indicating which y axis is being used for which curve.
- In the first paragraph of the introduction, the mentions of "single molecule dendra-2 actin" and "septin filament organization" seem out of place alongside the other much more general phenomena (especially the former, which should be deleted altogether).
- The statement in line 57 that "...the molecular orientation over a time window is defined by fluorescence lifetime, which..." is confusing and should be reworded for clarity, since molecular orientation and fluorescence lifetime are not related in this way. The lifetime defines a time window and the molecular orientation is what is probed within the time window.
- The comment in line 159-161 regarding sparsity requirements and their prevalence in the real world should be contextualized with some specific numbers.
- It is strange that "room temperature" is 285 Kelvin in line 229. Perhaps this is a typo?
- It is unclear to me whether the nonuniformity in Figs. 2n is related only to low SNR as seemingly claimed in the text. Since the plotted lifetime clearly correlates with the mask much more strongly than in other measurements. Although I don't think this substantially undermines the results, a comment rationalizing this observation is warranted.
- I found the overlay in Fig. 2b of the LIF signal and flame luminosity unclear. I think showing the panels side-by-side would be more clear since it is (perhaps) hard to distinguish dark blue overlaid with white from light blue overlaid with black.

With these comments addressed, I would be happy to support publication of this work in Nature Communications.

(Remarks on code availability)

I was able to run the code and obtain reasonable outputs. The code is sufficiently readable.

Version 1:

Reviewer comments:

Reviewer #1

(Remarks to the Author)
Please see the attached file.

(Remarks on code availability)

Reviewer #2

(Remarks to the Author)
I have no further comments, the paper is ready to be published.

(Remarks on code availability)

Reviewer #3

(Remarks to the Author)
The authors have made corresponding changes and the paper can be accepted.

(Remarks on code availability)
The authors have made corresponding changes and the paper can be accepted.

Reviewer #4

(Remarks to the Author)
My concerns have been adequately addressed and I am happy to recommend this manuscript for publication.

(Remarks on code availability)

Response to Reviewers

We thank the reviewers for their insightful comments, which have helped us improve the quality of our manuscript. Below, we provide point-by-point responses to all the comments (shown in blue). The changes in the text are highlighted in red.

Reviewer 1

Comment 1.0. This paper introduces a compressed ultrafast planar polarization anisotropy imaging (CUP2AI) technique based on compressive sensing theory, molecular rotational diffusivity, and femtosecond laser-sheet illumination to enable in-situ, real-time, non-invasive, wide-field anisotropy measurements in both liquid and gaseous environments. The study presents both single- and two-photon excitation setups.

Mapping molecular size is critical for understanding dynamic processes, particularly at imaging speeds reaching hundreds of billions of frames per second. The work is original and obtained high-resolution mapping of molecules size is of high significance.

[Response]: We thank the reviewer for the positive comments.

Comment 1.1. The paper presents various setups for different molecule types across two environments: liquid versus gaseous, single- versus two-photon excitation, single- versus double-cuvette, FL versus 4K versus 20K molecules, and at multiple imaging speeds. Consequently, it is crucial to improve the organisation of figures and tables. The Figures section, starting on page 19, is presented in an unstructured manner, making it difficult to discern the logic behind the arrangement of plots within each figure. Moreover, this logic often varies between figures, further complicating comprehension. For example, the caption in Fig. 2 describes the panels in the following order: (a)-(f), (k), (g), (l), (m), (h)-(j), (n), and (o). I strongly recommend a major reorganisation of the figures and their corresponding panels to enhance clarity. Further details are provided in the next comment.

[Response]: We agree with the reviewer that Fig. 2's caption does not follow the ordering of the panels in the figure. However, our Main Text on Pages 7 and 8 strictly follows the ordering of the panels. Therefore, in order to make everything consistent and enhance clarity, we added two sentences in Fig. 2's caption in Line 568 on Page 20: “**(f) – (j) Exemplary intensity, anisotropy,**

and lifetime results of single-cuvette experiments. (k) – (o) Exemplary intensity, anisotropy, and lifetime results of dual-cuvette experiments.”

Comment 1.2. The panels within each figure could follow a logical order, such as the processing streamline shown in Fig. S9. A natural structure for figures related to an experiment or setup might include: (i) setup, (ii) examples of E0, E1, and E2, (iii) reconstructed intensity evolutions, (iv) anisotropy dynamics, (v) anisotropy lifetime map, (vi) molecule volume map, and (vii) molecule size map. Given the multiple molecules and setups examined in this paper, the authors might consider focusing on a single representative molecule in the main text, while relegating similar (i)-(vii) style figures for the other molecules to the supplementary information. Alternatively, to allow comparisons between different molecules, the authors could adopt the suggested structure for columns within a figure. Each column would then contain panels (i)-(vii) for one molecule, enabling clearer comparison. Currently, the figures, with their coverage of multiple setups and molecules, lack coherence, which reduces their effectiveness. A dedicated figure could be used to compare only the final results, e.g., molecule size maps, across different molecules and/or setups. This suggestion is intended to emphasise the importance of improving figure and panel organisation.

[Response]: We totally understand the reviewer’s concern and thank the reviewer for this valuable suggestion. However, we’d like to argue that first of all, since there is too much data to show, Figs. 2 – 6 only contain selected results and the rest of results are given in Supplementary Figures. Second, the orderings of the panels in these figures strictly follow the logical flow of our Main Text. We believe that it helps the readers best understand our paper’s contents using the current method of presentation. Please also refer to our detailed responses to comments relevant to figures: Comment 1.13, Comment 1.14, Comment 1.18, Comment 1.19, Comment 1.22, and Comment 1.23.

Comment 1.3. The “Image Formation Model” section requires major revision. Multiple symbols are introduced for a single variable, which leads to unnecessary complexity and makes the equations difficult to follow. In addition, the mathematical derivations in the “Fluorescence

Anisotropy and Lifetime Extraction” section are not justified, or at least the underlying assumptions and steps are poorly explained. These need to be clarified and better justified to ensure the derivations are both accurate and understandable.

[Response]: We thank the reviewer for this insightful suggestion. We have modified our mathematical expressions and derivations according to the reviewer’s comments. Please refer to our responses to Comment 1.27, Comment 1.28, Comment 1.29, Comment 1.30, Comment 1.31, Comment 1.32, Comment 1.34, Comment 1.36, Comment 1.39, Comment 1.40, and Comment 1.41 for more details.

Comment 1.4. This research is not reproducible, as the authors explicitly state, “We have opted not to make the computer code publicly available because the code is proprietary and used for other projects.” I encountered issues reviewing the code provided on Codeocean, as it lacks proper comments and definitions for parameters. There are parameters used in the code that are not described in the paper. Additionally, the TwIST algorithm is presented as a black box, with no explanation of how the proposed regularization is incorporated into the algorithm. Furthermore, the convergence criteria for the TwIST algorithm, as applied to the minimization problem in the paper, are not discussed. Critical details of the deconvolution step, such as the algorithm used and the hyperparameters chosen, are also missing.

[Response]: We respectfully disagree with the reviewer. The Data Availability Statement is only used to state the availability of data. It is not supposed to be used to judge whether any research is reproducible or not. Besides, the data availability statement in our original manuscript was written before we were directed to upload the example codes in Code Ocean. Therefore, we need to revise our Data Availability Statement in the newer version: “**Example data processing codes can be accessed via Code Ocean.**” To comply with the editorial policy, we also added sentences on Page 6: “**A package containing example data and the processing code is publicly available in Code Ocean. It contains the major steps of obtaining molecule size maps from intensity movies.**” Please also see the response to Comment 1.12. In addition, our uploaded example code has been run and checked by other reviewers with no issues. To further address the reviewer’s concerns, we added more comments to the shared code to explain the parameters and functions.

Regarding the TwIST algorithm used in our work, please refer to our responses to Comment 1.32, Comment 1.34, Comment 1.35, Comment 1.36, Comment 1.37, and Comment 1.38 for more details.

Comment 1.5. Page 4: Line 105: “Our results demonstrate excellent agreement with existing literature and theory.” Could the authors clarify this claim? Which specific literature or theoretical results do their findings align with? Please provide references or further details to support this statement.

[Response]: We agree with the reviewer that the statement was not sufficiently clear. For the soot precursor molecules our results with the values of 15~45 Angstrom align excellently with typical literature values in a range from approx. 1~5 nm as shown in the review paper by Martin et al. (<https://doi.org/10.1016/j.pecs.2021.100956>) or dedicated studies, e.g., Desgroux et al. (<https://doi.org/10.1016/j.jaerosci.2024.106385>). For the FITC-dextran molecules, not all measurements perfectly align with previous studies (e.g., the work by Pu et al.), however, we discussed possible reasons for these deviations in the upcoming sections of the manuscript.

We rephrased the statement accordingly and added citations accordingly: “**Our results demonstrate excellent agreement to previous studies for the flame measurements [32] with existing literature and overall agreement to the molecular sizes of dextran molecules [31].**”

Comment 1.6. Page 5: Line 131: How were the masks generated? What was the ratio of 0 to 1 pixels, and how was this ratio determined? Is this ratio significant for the results, and if so, how does it impact the overall outcome?

[Response]: We used computer-generated pseudo-random pattern as our encoding masks. The corresponding pseudo-code in MATLAB is:

```
>> C = rand(100) > 0.5;
```

In the mask, there are 50% 0-pixels and 50% 1-pixels, because we try to best balance the intensity between the two complementary encoding channels. For instance, if we use 30% ratio in

one channel, the other channel will have 70% ratio, making a 2.5× difference between their signal levels, which potentially make reconstruction more difficult.

We added a sentence in the Digital micro-mirror device section in Methods: “**Note that we use a pattern with 50% 1-code and 50% 0-code to balance the reflected light intensities in the two complementary encoding channels.**”

Comment 1.7. Page 6: Line 170: Details of numerical deconvolution are not explained.

[Response]: The details of numerical deconvolution are explained in Section 5, Figs. S6-S8 and Table S3 in Supplementary Information. This was mentioned in our previous manuscript. We clarified this information by changing the content in the bracket to “(see **Section 5**, Figs. S6–S8 and Table S3 in Supplementary Information **for details of numerical deconvolution**)”.

Comment 1.8. Page 6: Line 175: Is “1PF” defined?

[Response]: Before this occurrence, both 1PF and 2PF were first defined in the “System and principle of CUP2AI” section in Results & Discussion (on page 4, line 116).

Comment 1.9. Page 8: Line 236: What is the interest for single- vs. double-cuvette measurements? A discussion would improve the clarity.

[Response]: We thank the reviewer for this insightful question. Imaging a single-cuvette setup that only contains one type of molecule of the same size has limitations in revealing the capability of our technique. Simultaneous imaging of double cuvettes containing vastly different molecules sheds light on the dynamic range of our technique since multiple molecular size classes have to be resolved in a single shot. This comes closer to potential application scenarios like a spray, where it can be expected that variously sized molecules are present, and the technique must be capable of capturing these. We added one sentence on Page 7: “**The dual-cuvette imaging additionally highlights the large dynamics that our technique can accommodate as multiple diversely sized molecules can be resolved in a single shot.**” Additionally, the experiment reveals that extinction

or attenuation effects of the laser pulse are of minor importance in terms of sizing molecules via anisotropy lifetime (also see response to Comment 3.2)

Comment 1.10. Page 8: Line 239: “This observation ... uniformly distributed size maps”: It is noted that the size maps are not uniform for 4K molecules. Could the authors explain why this is the case? Although it is mentioned later that “The results from 4K molecules have greater spatial non-uniformity owing to worse signal-to-noise ratio (SNR) from the sample’s lower fluorescence-intensity (see Figs. 2c-e).”, Fig. 2e shows that the signal intensity for 20K molecules is comparable to or lower than that for 4K molecules. Please clarify this discrepancy.

[Response]: We apologize for this misunderstanding. The molecule size maps of 4K molecules have worse spatial uniformity (equivalently more spatial variation) due to lower fluorescence signal levels. Figures 2c-2e may not be good results to demonstrate this difference since they are all normalized intensity plots, as already indicated in both figure captions (Page 20, Line 566 in the original manuscript) and Main Text (Page 7, Line 201 in the original manuscript). However, this difference in fluorescence intensity was very clearly observed in our experiments. Actually, the reviewer may want to take a closer look at the inset photos in Figs. 2c-2e, in which the fluorescence looks greenish for the 4K molecule (in Fig. 2d), while the fluorescence looks whiteish for the FL and 20K molecules (in Figs. 2c and 2e) due to saturation (much higher signals). To clarify, we changed Page 8, Line 243 to be “(see the inset photos in Figs. 2c-e, taken using the same exposure condition)” on in our revised manuscript. We also added the same description in Fig. 2’s caption in Line 568 on Page 20: “captured using the same exposure parameters.”

Comment 1.11. Page 9: Line 258: “the additional plots describing the remaining results are given in Figs. S11–S15 in Supplementary Information.”, For improved clarity, it would be helpful to specify which additional results are provided in the supplementary document.

[Response]: We provided more details on additional results in Line 248 on Page 9 of the revised manuscript: “(spatio-temporal dynamics of intensity and anisotropy, 2D maps of anisotropy and molecule size)”.

Comment 1.12. Page 15: Line 442: The authors mentioned that “The reconstruction algorithm is described in detail in Supplementary Information. We have opted not to make the computer code publicly available because the code is proprietary and used for other projects.”, however, important details of the reconstruction are missing and the results are not reproducible.

[Response]: Please see our response to Comments 1.4. We added more details of the reconstruction algorithm as suggested by the reviewer to address the reproducibility concern. However, we respectfully disagree with the reviewer since our results are reproducible, as can be proved by running the example codes that we have uploaded to Code Ocean, requested by the editorial policy of Nature Communications. In addition, we updated our Code Availability statement to “**Example data and processing codes can be accessed via Code Ocean.**” Regarding the TwIST algorithm used in our work, please refer to our responses to Comment 1.32, Comment 1.34, Comment 1.35, Comment 1.36, Comment 1.37, and Comment 1.38 for more details.

Comment 1.13. Page 19: Fig. 1: While the text explains the working principles of some elements used in CUP2AI, it does not cover others, such as BBO, HWP, SL, CyL, and SPF. Please provide explanations for these missing elements.

[Response]: A careful reading of our Main Text between Line 110, Page 4 and Line 137, Page 5 can identify the descriptions of the working principles of these elements.

- (1) Page 4, Line 114: “The laser beam was sent through a nonlinear crystal (BBO) to generate 400-nm second harmonic (SH).”
- (2) Page 4, Line 119: “After CyL, a half-wave plate (HWP) and a linear polarizer (P0) adjust the laser fluence while maintaining a linear polarization along the y-axis (blue arrow in Fig. 1a).”
- (3) Page 5, Line 132: “A high-numerical-aperture stereoscopic lens (SL) has dual functions: projecting the image to the DMD and collecting the two masked beams reflected from the DMD.”
- (4) Page 4, Line 117: “The laser beam has a $1/e^2$ diameter of about 10 mm and passes through a long-focal-length cylindrical lens (CyL) to form a laser sheet for intersecting a thin x - y plane inside the sample (see Fig. 1b).”

(5) Page 4, Line 115: “The 400-nm SH and the residual 800-nm pulses are utilized to excite 1P and 2P fluorescence (1PF and 2PF), respectively, selected by two interchangeable short-pass and long-pass spectral filters.” We added “SPF” in the revised manuscript: “... selected by two interchangeable short-pass and long-pass spectral filters (SPF and LPF).”

Comment 1.14. Page 20: The figure contains too much information, making it challenging for readers to follow. It needs to be modified.

[Response]: We agree with the reviewer that Fig. 2 has many plots, containing lots of information. However, all these plots are referred to in our Main Text. We think that all these plots are quite useful in explaining our system, experiments, and our results. And this information is helpful for readers to understand and appreciate our technique. Additionally, these plots are arranged in a manner that follows the logic flow in the Main Text, which makes sense since it is common for readers to refer to plots while reading the texts.

Comment 1.15. Page 20: What does negative anisotropy value mean?

[Response]: The definition of anisotropy can be found in Equation (10) (see Page 32, Line 732 in the original manuscript). When the parallel component of emission light I^{\parallel} (polarization in the same direction as the excitation pulse) has smaller intensity than that of the perpendicular component of emission light I^{\perp} (polarization direction orthogonal to the excitation pulse), the anisotropy a has negative value, and vice versa. The negative values in anisotropy occur at the tails of the fluorescence decay (see example plots in Figs. 2c-e), where the anisotropy tends to converge to zero ground (isotropic emission). The intensity from both polarizations are quite close to each other, it is reasonable to observe both small positive values and small negative values around zero from an experimentally acquired dataset due to noises.

Comment 1.16. Page 20: What does negative time values mean?

[Response]: We thank the reviewer for pointing this out. Time 0 in our plots represents the time of fluorescence excitation (typically the peak of fluorescence emission evolution), therefore

negative time means time before excitation. The finite rise time in negative time is due to the finite temporal resolution (finite width of temporal PSF or temporal impulse response) of the CUP2AI system. The experimentally measured temporal curve on the rising edge is essentially a convolution of an abrupt rise with the temporal PSF. For clarification, we define time 0 by adding a sentence in Line 202 on Page 7 of the original manuscript: “Time 0 on the x-axis is the time of fluorescence excitation.”

Comment 1.17. Page 20: What could explain non-uniform anisotropy map for 4K molecules?

[Response]: The non-uniformity in the 4K molecule’s anisotropy lifetime map is explained on Page 8, Line 241: “The results from 4K molecules have greater spatial non-uniformity owing to worse signal-to-noise ratio (SNR) from the sample’s lower fluorescence intensity”. Although in theory the 4K molecule should have uniform anisotropy lifetime distribution over space, the poorer SNR in the raw data, caused by lower quantum efficiency of the 4K molecules, leads to stronger variation in lifetime fitting, leading to non-uniformity. For more clarity, we added “...from the sample’s lower fluorescence intensity caused by its lower quantum efficiency” in Line 243 on Page 8 of the original manuscript.

Comment 1.18. Page 20: Right insets in (c)-(e) do not say much, due to their small sizes.

[Response]: Please refer to our response to Comment 1.10 above. The inset photos in Figs. 2c-e are useful to showcase the difference in fluorescence light levels between 4K (inset of Fig. 2d) and FL (20K) molecules (insets of Figs. 2c and 2e). Under the same exposure condition, the 4K photo shows greenish image of fluorescence while the FL and 20K photos show whiteish image due to saturation from very high intensity. This observation helps to explain the worse spatial uniformity present in 4K anisotropy lifetime (and molecule size) maps.

Comment 1.19. Page 22: The figure needs improvement as the ordering of panels is not intuitive. According to the data processing streamline in Fig. S9, panels (e) and (f) should precede (a) and (b). Consider organising the figure with one column per molecule, with molecule names as column

headers. Using a single color bar per molecule would better represent the dynamic range of molecule volume and size. In this case, panels (c) and (d) could be omitted, and instead, a single value indicating the variance across multiple experiments could be reported in the caption. Additionally, separate fitted anisotropy curves (as shown in panel (e)) could be presented for each molecule in a column-wise format as described.

[Response]: We thank the reviewer for this suggestion; however, we'd like to emphasize that the order of the panels in the figure follows the logical flow of our Main Text.

- (1) Panels (e) and (f) are used to explain the difference between mono-exponential and bi-exponential fit of the measured fluorescence anisotropy decay curves. We also use these plots to explain why different fit models are used for molecules of different sizes. Therefore, there is no need to show exponential fit for all molecules and all cases. An exemplary comparison, like the one in panel (e), is sufficient. In addition, in our Main Text, this discussion comes after the description of all our results, therefore, it makes sense to make it the last in the figure.
- (2) The plots in panels (c) and (d) contain results beyond those shown in panels (a) and (b), since there are three trial experiments for each sample, a total of 15 sets of data acquisition. However, panels in (a) and (b) are 5 examples from these 5 samples. Therefore, it makes sense to give summary plots like (c) and (d).
- (3) Based on the reasons above, we think it does not benefit our manuscript if we organize the figure in the way the reviewer suggested, which does not follow the logic of our write-up.
- (4) Additionally, we assign one colormap for all molecular volumes and a different colormap for all molecule sizes, so that we can easily make visual comparison among all samples.

Comment 1.20. Page 22: the arrows point to the directions of dipole moments”: how were they computed?

[Response]: The directions of these arrows in Fig. 3e are not quantitatively calculated, they are simply drawn to qualitatively represent the direction of molecular orientation over the evolution of its fluorescence emission. Larger molecules (like 20K) take much longer time to re-orient and thus a longer anisotropy decay time than those of smaller molecules (like FL). We added a sentence

in the figure caption in Line 591 on Page 22: “The arrow directions are qualitatively drawn for illustration only.”

Comment 1.21. Page 22: Is R^2 coefficient defined?

[Response]: Based on the theory of statistics, R^2 is referred as to “goodness of fit” or the “coefficient of determination”, which indicates how well a regression model fits the data. A higher R^2 value signifies a better fit to the data, with a perfect fit being represented by an R^2 of 1. This parameter is well known and widely accepted in the science and engineering community; therefore, we decided it is unnecessary to provide its full definition. We already mentioned “goodness-of-fit (R^2)” in Line 268 on Page 9 and “coefficient of determination (R^2)” in Line 594 on Page 22 of the original manuscript. We think these are sufficient for the readers. However, the reviewer may find its detailed definition in this link: https://en.wikipedia.org/wiki/Coefficient_of_determination.

Comment 1.22. Page 26: The style and ordering of the panels and subpanels need to be improved for better readability.

[Response]: We thank the reviewer for this suggestion. However, we’d like to mention that similar with other figures, the ordering of the panels follows the logic flow in our Main Text, which makes it easier for readers to follow. Meanwhile, we arranged multiple panels in Fig. 5 for straightforward visual comparison. Hence, we would prefer to keep our original style and ordering of the panels in this figure.

Comment 1.23. Page 28: The style of this figure is relatively better than the previous ones. However, in the first row from left to right, the reader encounters panels (a), (b), and (o), whereas they would naturally expect to see panels (a), (b), and (c). Including the names of the molecules as column headers would improve clarity. Additionally, are panels (a) and (b) different from those in Fig. 2? If so, please clarify the differences.

[Response]: We understand the reviewer’s confusion and concern regarding this figure. However, we arrange the figures in a way that follows the logic flow in the Main Text, since we think that

the readers will most likely refer to the figure when they read through the texts of the manuscript. In addition, we need to arrange the panels so that everything is presented in a clear way while saving space. Figures 6a and 6b are different from Figs. 2a and 2b, since we replaced the short-pass spectral filter (SPF) in Fig. 2 with a long-pass spectral filter (LPF) in Fig. 6. In Fig. 2, a SPF was used to select the 400-nm SH light for single-photon fluorescence, while in Fig. 6, an LPF was used to select the 800-nm light for two-photon fluorescence. For clarification, we added one sentence in the figure caption in Line 643 on Page 28: “In (a) and (b), a long-pass filter (LPF) is employed to select the 800-nm light for two-photon fluorescence, replacing the short-pass filter (SPF) used in one-photon fluorescence in Figs. 2a and 2b.”

Comment 1.24. Page 29: Line 660: What is the relation between sweeping voltage and the displacement between adjacent electrons?

[Response]: The exact relation between the sweeping voltage applied and the displacement between adjacent electrons is determined by the streak camera’s manufacturer (Hamamatsu), which is not accessible to us. A qualitative representation is given in the inset plot of Fig. S5a. However, we are able to select different options for sweeping speeds, which are pre-set by the manufacturer. The rule of thumb is that for electrons differed by a certain time gap, the faster the sweeping voltage the larger displacement (in the shearing direction) between these electrons (equivalently faster imaging speed for CUP2AI).

Comment 1.25. Page 29: Line 669: “diverse applications in modern optics” needs citation(s).

[Response]: We added 3 citations on Page 18 as references 41 – 43 to showcase the diverse applications of DMD in modern optics: (1) “Jeffrey B. Sampsel, *Digital micromirror device and its application to projection displays*. J. Vac. Sci. Technol. B, 1994, 12(6): 3242–3246.” (2) “Hakki H. Refai, James J. Sluss Jr., and Monte P. Tull, *Digital micromirror device for optical scanning applications*. Opt. Eng., 2007. 46(8) 085401.” (3) “Ayoub, A.B., and Psaltis, D. *High speed, complex wavefront shaping using the digital micro-mirror device*. Sci. Rep., 2021. 11, 18837.” They are cited in Line 669 on Page 29: “...in modern optics [41-43].”

Comment 1.26. Page 29: Line 675: The motivation for binning DMD pixels is missing.

[Response]: We thank the reviewer for pointing this out. We added more explanation in Line 676 on Page 29: “In compressed-sensing-based imaging, this encoding size (i.e. binning of DMD elements) is proportional to the spatial and temporal resolutions [43]. Therefore, a smaller binning factor is desired for a higher resolution. However, the SNR and space-charge effect of the streak camera compromise the contrast of the images when a small DMD binning is used. As a result, we need to trade off our choice to a sweet spot of 4×4 binning. A similar study on another streak camera can be found in the previous work [27].”

Comment 1.27. Page 30: Line 689: Why do the authors need to introduce both I^{\parallel} and I^{y-pol} ? Same for I^{\perp} and I^{x-pol} ; \mathbf{P}_1 and \mathbf{P}^{\parallel} ; and \mathbf{P}_2 and \mathbf{P}^{\perp} .

[Response]: We agree with the reviewer that these definitions are redundant. We simplified Equations (1) and (2) by using only I^{\parallel} , I^{\perp} , \mathbf{P}^{\parallel} and \mathbf{P}^{\perp} , which are more accurate. In addition, we revised the texts in Line 689 on Page 30: “In Eq. (1), I^{\parallel} and I^{\perp} stand for polarizations parallel (y -direction) and perpendicular (x -direction) to that of the excitation laser pulse, respectively.” We also revised texts in Line 692 on Page 30: “... \mathbf{P}^{\parallel} and \mathbf{P}^{\perp} are the operators of the two polarizers, \mathbf{P}_1 and \mathbf{P}_2 , whose transmission axes are in the y -direction and the x -direction, respectively,...”

Comment 1.28. Page 30: Line 700: It is not mentioned in the text what elements in streak camera correspond to the \mathbf{F}_0 , \mathbf{F}_1 , \mathbf{F}_2 , \mathbf{D}_1 , and \mathbf{D}_2 operations. For \mathbf{D}_1 and \mathbf{D}_2 it is somehow implicit. Do these symbols denote matrices or operators?

[Response]: These are operators on the images, corresponding to low-pass filters (\mathbf{F}_0 , \mathbf{F}_1 , \mathbf{F}_2) and distortion effect (\mathbf{D}_1 , \mathbf{D}_2) caused by the optical components in the imaging system. We added a sentence in Line 703 on Page 30: “Here, low-pass filtering, owing to the finite aperture and wavefront aberrations of the imaging optics, blurs the image, while geometric distortion, caused by aberration and slight misalignment, warps and rotates the image.”

Comment 1.29. Page 30: Equation (3) does not mathematically sound.

[Response]: We thank the reviewer for pointing this out. We modified the equation to:

$$E_0 = \mathbf{TF}_0 I^\parallel + \mathbf{TF}_0 I^\perp \quad (3.1)$$

$$\alpha_1 E_1 = \mathbf{TSP}_1 \mathbf{D}_1 \mathbf{F}_1 \mathbf{C}_1 I^\parallel \quad (3.2)$$

$$\alpha_2 E_2 = \mathbf{TSP}_2 \mathbf{D}_2 \mathbf{F}_2 \mathbf{C}_2 I^\perp \quad (3.3)$$

And we revised Line 703 on Page 30: “Eqs. (3.1) – (3.3) are re-written in the matrix form.”

Comment 1.30. Page 30: Eq. (4): It seems there is an \mathbf{F}_0 missing after \mathbf{C}_1 and \mathbf{C}_2 .

[Response]: The low-pass filtering operators for the two time-shearing views are \mathbf{F}_1 and \mathbf{F}_2 , which come after \mathbf{C}_1 and \mathbf{C}_2 , respectively. These two operators were correctly placed in Eq. (4).

Comment 1.31. Page 30: Eq. (4) is true if $\mathbf{C}_1 \mathbf{F}_0$ and $\mathbf{C}_2 \mathbf{F}_0$ do not change the polarisation, which seems to be fine, but it is worth mentioning.

[Response]: We agree with the reviewer and added this information in the updated manuscript in Line 706 on Page 31: “Note that none of the operators in Eq. (4) changes the polarization state of light.”

Comment 1.32. Page 30: Eq. (5): The properties of the matrix \mathbf{O} are not discussed. Does it hold any mathematical property, e.g., full rank? Does the convergence of the TwIST algorithm depend on the properties of this matrix?

[Response]: The matrix \mathbf{O} contains all the operators in the optical imaging system, meaning $\mathbf{O} =$

$$\begin{bmatrix} \mathbf{TF}_0 & \mathbf{TF}_0 \\ \mathbf{TSD}_1 \mathbf{F}_1 \mathbf{C}_1 & \mathbf{0} \\ \mathbf{0} & \mathbf{TSD}_2 \mathbf{F}_2 \mathbf{C}_2 \end{bmatrix}. \text{ In an ideal scenario in compressed sensing, matrix } \mathbf{O} \text{ is full rank, however, it}$$

may not be full rank in practice. The regularization algorithm TwIST works best when the matrix \mathbf{O} is close to being full rank. A major contributor to the properties of \mathbf{O} is the selection of the encoding mask $\mathbf{C}_1 = \mathbf{1} - \mathbf{C}_2$. In this study, we chose to use pseudo-random masks, which led to reasonable convergence and

fidelity in reconstruction. We acknowledge that alternative mask designs can lead to similar or even better results, which may be a future direction worth investigation.

Comment 1.33. Page 30: Line 710: For being self-contained, a summary of calibration procedure must be given in the text.

[Response]: We agree with the reviewer and added a summary of calibration procedure in Line 709 on Page 30: “To calibrate for F_1 , C_1 , F_2 , and C_2 , we took an image of the encoding masks when the imaging system was flood illuminated by a uniform beam (collimated green laser diode) and the sweeping voltage in the streak camera was turned off (operated as a conventional camera). Using the same experimental configuration, regular patterns (e.g. checkerboard or dot array) were then imaged to calibrate for the distortions D_1 and D_2 . The temporal shearing operations (S) for different imaging speeds are provided by the streak camera manufacturer.”

Comment 1.34. Page 30: Eq. (6) does not mathematically sound. The symbol I should not denote both the target intensity and an optimisation variable. It may be replaced by another symbol in the minimisation problem formula.

[Response]: We thank the reviewer for pointing this inconsistency out. In the updated Equation (6), we removed \hat{I} , which is unnecessary. The remaining of the equation serves our purpose fine.

Comment 1.35. Page 30: Line 716: “The regularizer promotes sparsity in the solution and assists reconstruction to converge to global minimum. ” This is true only if the optimisation problem is convex, otherwise there will be local minima issue. Have authors investigated the convexity of minimisation problem in Eq. (6)?

[Response]: The minimization problem defined in Eq. (6) is convex optimization, which has been extensively investigated in Coded Aperture Snapshot Spectral Imaging (CASSI) technique. The image reconstruction (optimization) algorithm used in this work resembles that extensively utilized in CASSI. The convexity of the optimization algorithm can be found in reference papers:

- (1) G. R. Arce, D. J. Brady, L. Carin, H. Arguello and D. S. Kittle, *Compressive Coded Aperture Spectral Imaging: An Introduction*, IEEE Signal Processing Magazine, vol. 31, no. 1, pp. 105-115, Jan. 2014.
- (2) Patrick Llull, Xuejun Liao, Xin Yuan, Jianbo Yang, David Kittle, Lawrence Carin, Guillermo Sapiro, and David J. Brady, *Coded aperture compressive temporal imaging*, Opt. Express 21, 10526-10545 (2013).
- (3) Kerkil Choi, David J. Brady, *Coded aperture computed tomography*, Proc. SPIE 7468, Adaptive Coded Aperture Imaging, Non-Imaging, and Unconventional Imaging Sensor Systems, 74680B (24 August 2009).

We added these references as Refs [45] [46] [47] in the updated manuscript.

Comment 1.36. Page 30: Line 720: $\Phi(\mathbf{I}) = \Phi(\mathbf{I}^{\parallel}) + \Phi(\mathbf{I}^{\perp})$ is confusing. The authors may use different symbol for the right-hand-side.

[Response]: We agree with the reviewer and replaced the symbols on the right-hand-side Φ by Ψ .

Comment 1.37. Page 30: Eq. (7): Could the authors provide a reference for the type of the TV function used here? For a 3D TV regularisation this is not a natural choice.

[Response]: The use of total variation (TV) as regularizer in optimization problem defined by Eq. (6) is a popular choice. To cite a few reference papers using TV regularizer from the CASSI technique:

- (1) Patrick Llull, Xuejun Liao, Xin Yuan, Jianbo Yang, David Kittle, Lawrence Carin, Guillermo Sapiro, and David J. Brady, *Coded aperture compressive temporal imaging*, Opt. Express 21, 10526-10545 (2013).
- (2) Kerkil Choi, David J. Brady, *Coded aperture computed tomography*, Proc. SPIE 7468, Adaptive Coded Aperture Imaging, Non-Imaging, and Unconventional Imaging Sensor Systems, 74680B (24 August 2009).
- (3) David Kittle, Kerkil Choi, Ashwin Wagadarikar, and David J. Brady, *Multiframe image estimation for coded aperture snapshot spectral imagers*, Appl. Opt. 49, 6824-6833 (2010).

In addition, we also described the use of TV regularizer in our previous works, such as Refs [22] and [23]. We added these citations as Refs [46] [47] [48] to validate the implementation of Eq. (7). A sentence is also added accordingly on Page 33: “The TwIST algorithm and the use of total variation as the regularizer have been extensively implemented in the previous Coded Aperture Snapshot Spectral Imaging (CASSI) technique [46-48].”

Comment 1.38. Line 722: Have the authors used TwIST algorithm as a black-box? It is not clear how the proposed TV regularisation was handled by TwIST algorithm. How did they set the hyperparameters? What halting criteria was used? Does the problem in Eq. (6) satisfy the convergence conditions of TwIST algorithm?

[Response]: TwIST algorithm, first proposed in 2007 in Ref [40], has been extensively applied in compressed-sensing-based computational imaging techniques. One prominent example is Coded Aperture Snapshot Spectral Imaging (CASSI), in which TwIST algorithm has been successfully implemented (see the newly added Refs [44] – [48]). We added a sentence in Line 722: “The TwIST algorithm and the use of total variation as regularizer have been extensively implemented in the previous Coded Aperture Snapshot Spectral Imaging (CASSI) technique [46-48].”

To adopt the TwIST algorithm for our work, we need to give the raw images (from both streak camera and the external camera) and the calibration data containing operators in matrix \mathbf{O} as inputs. In addition, we need to modify the regularizer function to total variation based on its defined in Eqs. (7.1) and (7.2).

As pointed out by the reviewer, it is necessary to set the proper parameters for the reconstruction algorithm. There are two kinds of parameters we optimized: one is the regularization parameter ω in Eq. (6), and the other two are the weighting factors α_1 and α_2 in Eq. (4). For parameter optimization of every set of acquired data, we employed a brute-force technique, in which we scanned over wide ranges for all these parameters and found one combination offering the best solution. The typical optimal value of the regularization parameter ω is around 10^{-4} , and the optimal α_1 and α_2 typically falls between 0.1 and 0.5. We applied two stop criteria: one is the total number of iterations, which is set to be 100 in this work; the other is the minimum improvement allowed after one iteration, which is about 10^{-3} , depending on the objective function used (defined by Eq. (6)). To provide more information, we added sentences in Line 723: “In this

current work, both raw images (captured by the streak camera and the external camera) and the calibrated operators in matrix \mathbf{O} are inputs to the TwIST algorithm. The regularizer function needs to be modified according to TV's definition in Eqs. (7.1) and (7.2). The regularization parameter ω and the weighting factors α_1, α_2 are optimized and selected based on a brute-force searching step in which wide parameter spaces are scanned. The ω parameter is typically around 10^{-4} , while α_1 and α_2 fall between 0.1 and 0.5. Two stop criteria are employed: (1) the maximum number of iterations is 100; (2) the minimum improvement in the objective function (defined in Eq. (6)) after one iteration is set to be 10^{-3} . The reconstruction algorithm terminates when either criterion is satisfied.”

Please refer to our response to Comment 1.35 above for the convergence and convexity of the minimization problem described in Eq. (6).

Comment 1.39. It is not clear how Eq. (9) and the factor of 2 within it were derived.

[Response]: The reason for factor 2 was laid out in detail in multiple publications (e.g. Pu et al. or Jameson and Ross <https://doi.org/10.1021/cr900267p>), which was cited as Ref [31] at the beginning of this section (above Eq. (8)). For the reviewer, the corresponding sections of that paper are copied below:

“Any random-oriented emission can be decomposed to have three projective components in the X, Y, and Z directions. One can assume that fluorescence intensities emitted from a dipole can be expressed by three components of $I_x, I_y,$ and $I_z.$ ”

“Referring to the polarization direction of excitation light, I_x is usually called as a parallel component, while I_y and I_z are called as perpendicular components. Since the excited molecules are symmetrical about the polarization direction of exciting light, it is reasonable to assume $I_y = I_z.$ Let $I_x = I^{\parallel}, I_y = I_z = I^{\perp}.$ ”

In our Eq (9), the total sum in the denominator therefore simplifies to the factor of 2. We don't think this needs to be repeatedly mentioned in our manuscript.

Comment 1.40. It is not clear why $a(t)$ in Eq. (10) follows an exponential decay function. Detailed derivation is needed for clarity.

[Response]: We thank the reviewer for this question. However, we refer the reviewer to the detailed derivation in the Pu et al. paper (Ref [31]). The length of the derivation is beyond the scope of our manuscript.

Comment 1.41. The development from Eq. (12) to Eqs. (13) and (14) is confusing. How accurate a mono-exponential decay assumption is? Could the authors provide a justification for obtaining Eq. (13) and (14)?

[Response]: The derivation of the fluorescence decay following a mono-exponential or multi-exponential decay is laid out in multiple textbooks, such as (1) Valeur, B. Molecular Fluorescence: Principles and Applications; Wiley-VCH: Weinheim, Germany, 2002; and (2) Lakowicz, J. R. Principles of Fluorescence Spectroscopy, 2nd ed.; Kluwer Academic: New York, 1999. We therefore refrain from reiterating these principles as the manuscript focuses on recent findings applying these techniques. We had additionally provided discussion in the manuscript about why we think mono- or bi-exponential are utilized (see pages 9-10).

Comment 1.42. Line 758: How did the authors obtain system's temporal response? a cross-reference to Sec. S5 is needed.

[Response]: We added a sentence in Line 758 to refer to that section in SI: **“Please see Section 5.1 in Supplementary Information for the calibrated temporal PSFs.”**

Comment 1.43. Line 759: The deconvolution process is not explained. Please provide details on the algorithm used and the parameter settings applied during deconvolution

[Response]: The numerical deconvolution method in this work has been detailed in Section 5.2 in Supplementary Information, supported by the plots in Fig. S8. We already referred to that section in our previous version. To address the reviewer's comment, in the updated manuscript, we added a new sentence in Line 760: **“To be specific, a look-up-table (i.e. a calculated plot) is used to infer deconvolved lifetime from the measured lifetime.”**

Comment 1.44. A justification is needed for obtaining Eq. (18) from Eq. (17).

[Response]: Equation (18) is simply derived from Eq (17) by solving for the diameter $d=2r$. Additionally, the dependencies on the pixel coordinates (x,y) are added in Eq (18) to highlight how the values in the image are obtained. We don't think this needs further justification.

Comment 1.45. Supplementary Information Page 2: What is the difference between the two dashed boxes in the top and bottom panels? The figure does not clearly illustrate the differences between the 1P and 2P systems.

[Response]: We think the figure clearly point out the difference between the dashed boxes in Figs. S1a and S1d. In Fig. S1a, the top one, we use a SPF to select the SH light (400 nm, represented by blue shades) to excite 1P fluorescence, while in Fig. S1d, the bottom one, we use an LPF to select the NIR light (800 nm, represented by the red shades) to excite 2P fluorescence.

Comment 1.46. Supplementary Information Page 8: Section 5.1 needs to be completed. Representative examples of raw data (before fitting) and the fitting errors should be provided. Additionally, are the reported temporal resolutions based on only one trial, or are they averaged over multiple trials?

[Response]: We thank the reviewer for this advice. We have updated Supplementary Fig. S6 by including the experimental measurement data, marked by squares of the same colors with the Gaussian-fit plots. The root-mean-square (RMS) of the fitting error is less than 2% for all cases. These are single-shot measurements and are not averaged over multiple trials. We added one sentence in Fig. S6's caption: "They are fits to Gaussian functions. The experimental measurements are plotted as squares in the same colors." We also added another sentence in Line 107 on Page 8 in Supplementary Information: "The root-mean-square (RMS) of the fitting error is less than 2% for all cases."

Comment 1.47. Supplementary Information Page 9: Line 120: I think it should be “after deconvolution”. item Line 123: Implemented deconvolution process needs to be explained.

[Response]: Please refer to our response to Comment 1.46 for the explanation of deconvolution process. Here, we respectfully disagree with the reviewer. Line 120 on Page 9 should be “after convolution” instead of “after deconvolution”. Here, we need to build the LUT for numerical deconvolution, so we need to do numerical convolutions to build up this dataset, which is essentially a corresponding relationship between convolved and deconvolved decay constants.

Comment 1.48. Supplementary Information Page 10: Plotting the $y = x$ curve would be helpful for understanding the impact of convolution.

[Response]: We thank the reviewer for this good suggestion. We updated Fig. S8 by adding this $y = x$ curve and added a sentence in the figure caption: “**The black dashed line is the unconvolved decay constant, used to manifest the impact of deconvolution.**”

Comment 1.49. Supplementary Information Page 10: It seems that the authors have obtained these curve from only one trial. However, a Monte-Carlo experiment and is important to indicate the stability of the method.

[Response]: The temporal PSF calibration data were taken in single shot, as the reviewer has suggested. Actually, we have done multiple trials of the calibration experiments, and the results look quite similar. This is understandable since the calibration sample is simply illuminating a binary pattern by an ultrashort laser pulse, which is supposed to be repeatable and stable. Therefore, we think that there is no need to conduct a Monte-Carlo experiment. For clarification, we added two sentences in Line 105 on Page 8 of the Supplementary Information: “**Since this simple ultrafast event is repeatable and stable, the results from multiple trials are similar. Therefore, we did not average over multiple shots for this calibration.**”

Comment 1.50. Supplementary Information Page 17: The style of Figs. S11 and S.15 are not consistent with the style of Figs. S12, S13, and S14 where the results of capture 1 are also shown.

[Response]: The inconsistency among these figures is because some of the data are already shown in Fig. 2 and Fig. 3 in the Main Text. The 1D, 2D, and 3D plots in Figs. S11 – S15 are just the remaining results that not included in the Main Text.

Reviewer 2

Comment 2.0. The author demonstrated the world's fastest fluorescence polarization anisotropy imaging using their streak camera. The analysis method is new and powerful, the physics and the data are profound and solid. I therefore support its publication in Nature Communications. I have only one comment:

[Response]: We thank the reviewer for the strong support.

Comment 2.1. The data acquisition is so fast so that for each molecule, there is even less than one photon detected in each frame. So, the result is still an ensemble average. What is the new information this ultrafast method can achieve when compared with those methods that capture the time-domain average signal (such as the method reported in ref. 12)? Since the average on the time domain is more or less equivalent to the average over many molecules (for an ergodic random process). The authors may need to explain this concern slightly.

[Response]: We respectfully disagree with the reviewer that the captured signals are temporally averaged. As can be taken from the movies, the frames can be captured from a single shot. The signal intensity on the photocathode of the streak camera relates to the photon flux of the fluorescence signal and the transmission characteristics of the detection path. Based on certain assumptions, we did an assessment to provide some plausible values:

Fluorescein concentration: 50 μM

Fluorescein absorption: $1.5 \times 10^4 \text{ M}^{-1} \text{ cm}^{-1}$

Quantum yield: 0.8

Laser wavelength: 400 nm

Laser fluence: 3.1 mJ/cm^2

Laser sheet width: 58 μm

Physical size of one pixel on object space: 84 μm

Limiting NA of optics: 12 mm / 200 mm = 0.06

Lens transmission: 0.92⁶

Mirror reflection: 0.95⁴

DMD efficiency: 0.5

Total number of frames: 300

This gives us

- a. For 1PF of fluorescein, we have 1.25×10^3 photons per frame per pixel

After conversion to other scenarios:

- a. For 1PF of 4K, we have 0.13×10^3 photons per frame per pixel
- b. For 2PF of fluorescein, we have 0.56×10^3 photons per frame per pixel
- c. For 2PF of 4K, we have 0.06×10^3 photons per frame per pixel
- d. For 1PF of PAH in flame, we have 0.10×10^3 photons per frame per pixel

We hope this sufficiently addresses the reviewer's concerns.

Reviewer 3

Comment 3.0. This work provided a new method of polarization anisotropy, named compressed ultrafast planar polarization anisotropy imaging (CUP2AI). The in-situ, transient and 2D measurement of molecule size both in liquid and gas phase is achieved by leveraging rotational diffusivity of molecules, femtosecond laser-sheet illumination, and compressed sensing. The CUP2AI provides a good evaluation of sizes of fluorescein in water and polycyclic aromatic hydrocarbons in flames. This work would benefit the chemical, biological and even engineering community.

[Response]: We thank the reviewer for the positive feedback and recognizing the potential impacts of our work.

Comment 3.1. The measurement is based on the model of Stokes-Einstein-Debye relation with the parameters of anisotropy decay time, temperature and viscosity. The decay time is fitted by the CUP2AI measurement result. Could you please evaluate the influence of parameter errors?

[Response]: We thank the reviewer for this insightful suggestion. We agree with the reviewer that an assessment of the impact of those parameters is important. We performed a parametric study (similar to the approach described in Bauer et al. <https://doi.org/10.1007/s00340-019-7219-7>) of volume V , viscosity η and temperature T based on a “synthetic measurement” (using the forward model and corrupting the resulting decay curve with a Gaussian noise of 5% mimicking the measurement noise). The underlying ground truth values were exemplary, set to typical values of $V = 3.0 \text{ k} \text{ \AA}^3$, $\eta = 1.234 \times 10^{-3} \text{ Pa s}$ and $T = 285 \text{ K}$. We next evaluate the decay curve assuming normal distribution for the uncertainties of all parameters and added reasonable prior assumptions on temperature (0.5 K) and viscosity ($0.025 \times 10^{-3} \text{ Pa s}$) of one sigma, respectively. The resulting normal distributed uncertainty in the key quantity of interest – volume, was found to be $0.1 \text{ k} \text{ \AA}^3$. For a fixed temperature of 285 K (which is assumed, but can be measured relatively precisely) the resulting correlation between viscosity and volume is shown below:

A full uncertainty quantification is beyond the scope of the paper and therefore we did not add it to the manuscript. Yet, we hope that our analysis can sufficiently address the reviewer's remark.

Comment 3.2. According to Fig. S2 and Fig. S3, the laser intensity is not uniform distribution among the 2D view. Does the laser intensity distribution cause the measurement error? Does the parameter of anisotropy decay time depend on the laser intensity?

[Response]: As a ratiometric method using simultaneous acquisition of both channels, our technique in theory does not depend on the laser intensity or spatial variations in the laser profile. However, certain aspects such as photobleaching or dye fragmentation may be caused by utilizing too strong laser intensities. Additionally, the SNR may be affected by a low signal intensity related to a low laser intensity. This is also mentioned and discussed in the comparison of the 1PF and 2PF measurements on Page 13.

Comment 3.3. The real molecules are not sphere. The results of CUP2AI provide the hydrodynamic diameter. What is the relationship between the real morphology and the diameter?

[Response]: Certainly, the conversion from a real molecular diameter to the hydrodynamic diameter demands the use of a shape conversion factor. Since our current manuscript limits itself to the determination of the hydrodynamic diameter, we however refrain from the use of such a conversion.

Comment 3.4. $a(x, y, t)$ is assumed to follow either a mono-exponential decay or a bi-exponential decay, according to Eq. (11)-(14). Why? In Fig. 2c and Fig. 2d, the anisotropy is fitted by mono exponential decay, but the anisotropy is fitted by bi-exponential decay in Fig. 2e. What is the reason for fitting by different functions?

[Response]: To explain why we need to use mono-exponential fit for FL and 4K molecules while using bi-exponential fit for 20K molecule, we'd like to cite the write-up from Line 262 to Line 270 on Page 9 of our original manuscript: "When the molecule is too large, it no longer possesses regular shape so that its anisotropy evolution deviates from a simple mono-exponential decay [31]. For large molecules (e.g. 20K), additional depolarization pathways caused by internal molecular motions start to take over, contributing to a shorter anisotropy decay than that caused by Brownian rotation. In this case, bi-exponential model is employed where the long decay constant is taken as the anisotropy lifetime used in sizing. See Methods and Eq. (14) for more descriptions. In Fig. 3f, the mono-exponential fit of 20K molecule clearly shows a worse goodness-of-fit (R^2) than the other two types of molecules, supporting its anisotropy's departure from mono-exponential decay, while the bi-exponential fit exhibit a significant improvement in R^2 (from 0.78 to 0.97)."

This phenomenon was also described in detail in a review paper, Ref [31]: "Pu, Y., et al., *Review of ultrafast fluorescence polarization spectroscopy [Invited]*. Applied Optics, 2013. 52(5): p. 917-929."

In fact, we already had a detailed explanation from Line 743 to Line 756 on Page 33 in our originally submitted manuscript [see right below Eq. (14)]. The reviewer may want to refer to those two paragraphs as well.

Comment 3.5. As for the PAH measurement in the flame, the non-premixed kerosene wick flame could be unstable. Do the authors try other stable flames? Do the authors compare the CUP2AI results and previous literature investigations?

[Response]: We thank the reviewer for this question. These measurements are single-shot, using only a single laser pulse, so flame instability is not a concern in our experiments. Whether the flame is stable or unstable, we can make accurate fluorescence anisotropy measurements. The

polarization anisotropy of similar flame aligns with our results and is included in the main manuscript (see Refs [13] and [36]).

Reviewer 4

Comment 4.0. In this manuscript, the authors present a new imaging method extending the technique of compressed ultrafast photography to spatially mapping polarization anisotropy by wide-field imaging of fluorescence generated by a femtosecond excitation pulse. Under some assumptions about the rotational dynamics of the emission dipole, polarization anisotropy measurements are translated to measurements of molecular size via determination of the anisotropy decay time. The new technique is characterized using proof-of-principle experiments with fluorophore solutions and a more interesting demonstration with laser-induced fluorescence from hydrocarbons formed in a kerosene flame.

The results presented in this work are novel and have clear application to nanoscale size measurement in a variety of experimental contexts. Two-dimensional characterization of transient species in flames and aerosols as done in this paper is of particular interest since this is a challenging experimental regime. The use of a single optical excitation pulse is also of special interest for light-sensitive environments.

The manuscript is impressively thorough in detailing the experimental design, analysis methodology, and results. The detailed schematics and parts list in the Supplementary Materials promotes transparency and reproducibility, and the extensive review of different aspects of the experiment throughout the text make it easy to follow exactly what the authors did.

[Response]: We thank the reviewer for acknowledging the novelty and significance of our work.

Comment 4.1. I found the ovals with arrows in e.g. Figs. 2, 6, S4, and S11-17 confusing. Please add a statement about them to the figure caption, or consider an alternative means of indicating which y axis is being used for which curve.

[Response]: We apologize for the confusion. For the purpose of clarification, we added a corresponding statement in Fig. 2's caption: "The circles and arrows in (c), (d), (e), and (m) group the plots for either the left-y axis or the right-y axis." Similar statements are added for all other figures: Fig. 6, Fig. S4, and Figs. S11-S17.

Comment 4.2. In the first paragraph of the introduction, the mentions of “single molecule dendra-2 actin” and “septin filament organization” seem out of place alongside the other much more general phenomena (especially the former, which should be deleted altogether).

[Response]: We agree with the reviewer and have replaced them with a general term “**cytoskeleton protein interactions**”, while retaining the references.

Comment 4.3. The statement in line 57 that “...the molecular orientation over a time window is defined by fluorescence lifetime, which...” is confusing and should be reworded for clarity, since molecular orientation and fluorescence lifetime are not related in this way. The lifetime defines a time window and the molecular orientation is what is probed within the time window.

[Response]: We agree with the reviewer that fluorescence lifetime does not directly define molecular orientation but rather reflects the time a molecule stays in an excited state before emitting a photon. Therefore, we have modified the sentence accordingly: “**From fluorescence lifetime measurements due to one-photon (1P) or two-photon (2P) excitation, time-resolved PA, i.e., the molecular orientation dynamics over a given time window, typically in the range of a few nanoseconds for frequently used fluorophores, can be inferred.**”

Comment 4.4. The comment in line 159-161 regarding sparsity requirements and their prevalence in the real world should be contextualized with some specific numbers.

[Response]: We thank the reviewer for this insightful suggestion. We actually characterized the data sparsity in a previous work (Ref [26]), in which we imaged a wide range of physical phenomena, including fluorescence samples. Fluorescence imaging has a data sparsity of around 95% (correspondingly, data density = 5%), and the average sparsity for all studied physical phenomena is ~96% (corresponding to data density = 4%). We added a sentence in Line 161 where sparsity is discussed: “**Our previous study showed a typical sparsity of ~95% for most ultrafast phenomena [26].**”

Comment 4.5. It is strange that “room temperature” in 285 Kelvin in line 229. Perhaps this is a typo?

[Response]: The lab room where our works were conducted was actually at 285 K. We kept it at this temperature to make sure our laser operates at its best condition.

Comment 4.6. It is unclear to me whether the nonuniformity in Figs. 2n is related only to low SNR as seemingly claimed in the text. Since the plotted lifetime clearly correlates with the mask much more strongly than in other measurements. Although I don't think this substantially undermines the results, a comment rationalizing this observation is warranted.

[Response]: We thank the reviewer for this sharp observation. The fact that the 4K molecule results correlates with the shape of the mask is truly due to low signal, thus low SNR.

First of all, there is some small amount of light transmitting through the dark region of the mask (leakage due to imperfect black ink used in printing). This is more obvious when we have more closely packed features such that those used in dual-cuvette experiments. This is the reason why the letters in the dual-cuvette experiments are not very well resolved and we can see a shape representing the outer edge wrapping around the letters. Secondly, the fluorescence intensity from FL and 20 K molecules is much stronger than that from 4K molecules. Therefore, we can still find the anisotropy lifetime very well even at the low-transmission (dark) regions of the mask between the letters (see the FL and 20K plots in Figs. 2n and 2o). On the other hand, for the 4K, we can still find the anisotropy lifetime accurately at the high-transmission (white) regions of the mask (letters); however, the anisotropy lifetime data extracted at the low-transmission (dark) regions of the mask (between letters) become inaccurate, since the signal level (also SNR) at those places are so much lower. This explains why the 4K sample shows a good resemblance to the mask.

We added a few sentences in Line 22 on Page 8 to explain this: “**Note that the lifetime maps do not resolve the letters well due to imperfect printing of the masks, allowing a small but detectable amount of light transmitting through the spacings between the letters. This is more obvious for FL and 20K molecules whose fluorescence emissions are stronger. However, for 4K molecules, the leaked light signals are too weak to estimate the anisotropy accurately. Therefore, the 4K molecular lifetime map (left side of Fig. 2n) resembles the printed mask more than the others.**”

Comment 4.7. I found the overlay in Fig. 3b of the LIF signal and flame luminosity unclear. I think showing the panels side-by-side would be more clear since it is (perhaps) hard to distinguish dark blue overlaid with white from light blue overlaid with black.

[Response]: Thank you for bringing this to our notice. We highlighted the LIF signal in blue in our measurements, as it is the sole signal of interest in this investigation. In addition, we'd like to use the overlay to illustrate the spatial location of LIF signals relative to flame luminosity. Placing the panels side-by-side may not show the relative position between LIF and luminosity well. Meanwhile, the profile of flame luminosity, which is also seen in the left panel of Fig. 4b, is not critical in our work, therefore, we used it as the background in this overlay.

Response to Reviewer

Reviewer 1

Overall, I am disappointed that the majority of my comments and concerns remain unaddressed in the manuscript. The authors have made insufficient effort to engage with the feedback provided during the previous round of revisions, and many of my comments have either been disregarded or inadequately addressed. For example, critical issues regarding the association of mathematical operators to physical components, the convergence of the reconstruction algorithm, and the clarity and arrangement of figures have been met with insufficient responses, and these concerns are not reflected in the revised text. As an applied mathematician with extensive experience in the theory of compressive sensing and the TwIST algorithm, I am particularly concerned about the authors' inadequate response to the questions about the algorithm's convergence. Based on their responses, I have new and serious concerns about the robustness and theoretical validity of their approach. The lack of clarity and justification for algorithmic convergence undermines the credibility of the reconstruction framework they propose.

Additionally, I note that the provided code does not include the reconstruction of the data but only post-processing steps. This omission raises concerns about the reproducibility of the results presented in the manuscript. In its current form, I do not believe the manuscript is ready for publication. The authors need to provide concrete answers to the questions posed during the previous round of review and address all comments in a substantive and transparent manner. Despite the significance and excellent novelty of the proposed framework, without these revisions, the paper falls short of the standards required for publication.

[Response]: We appreciate the reviewer's recognition of the novelty and importance of our proposed framework. We regret that our earlier revisions did not fully address all of the reviewer's concerns. It was never our intention to overlook any feedback, and we acknowledge the need for a further revision. In this revised version, we are committed to thoroughly addressing the remaining comments and ensuring they are adequately incorporated into the manuscript.

We understand the reviewer's concerns regarding the theoretical validity and robustness of our approach. To address these, we have provided a discussion of the convergence conditions for TwIST and outline the assumptions underlying our reconstruction framework.

In cases where certain theoretical aspects extend beyond the scope of this work, we have clearly identified them as open problems for future investigation. We have adapted the main text believe that our comments and changes further are sufficient for a positive outcome of our manuscript.

Please find our point-by-point response to the reviewer's comments.

Comment 1.4. I am unclear about what the authors disagree with! In the original manuscript, the authors explicitly stated that the code would not be made publicly available. As a reviewer, I am expected to evaluate the paper based on the provided text, and at that time, the statement regarding the code's unavailability was made in the "Code Availability Statement." Additionally, the claim that "The Data Availability Statement is only used to state the availability of data" seems misplaced since the issue pertains to the code availability, which directly impacts the reproducibility of the research.

[Response]: We would like to clarify that the reason for not making the code publicly available is related to ongoing patent considerations. We hope the reviewer can appreciate that the manuscript provides a state-of-the-art optical imaging method backed by sufficient detail for the findings to be understood and critically evaluated.

Comment 1.7. Based on the authors' response, this process appears to be a calibration of the decay time rather than a true deconvolution. Deconvolution is an inverse problem aimed at decoupling the physical system response from the sample system response, whereas the authors modify the decay time using a lookup table. This distinction should be clarified.

[Response]: We agree with the reviewer and changed the wording to "method used to approximate a deconvolution" in the according sections.

Comment 1.18. Thanks for the clarification. This explanation should be included in the text or figure caption for clarity.

[Response]: We have added the following sentences to the main text and the caption:

"The inset photos in Figs. 2c-e are used to showcase the differences in fluorescence light levels between 4K (inset of Fig. 2d) and FL and 20K molecules in the insets of Figs. 2c and 2e, respectively. Under the same exposure condition, the 4K photo shows greenish image of fluorescence while the FL and 20K photos show whiteish image due to saturation from very high intensity."

“Inset photos show the greenish image of fluorescence of 4K while whiteish images of FL and 20K due to higher intensity under the same exposure condition”

Comment 1.22. I would like to elaborate further. First, panels q, r, s, and t are not described in the text, which can confuse readers. Additionally, based on the current arrangement and labels in the figure, a reader is expected to follow an unconventional order: starting from panel a (top-left) to panel e (middle-left), then to panel i (top-right) and panel m (middle-right), before proceeding to panels q–t in the bottom row from left to right. A more natural ordering, which does not conflict with the logic of the text, would involve arranging the figure in rows from left to right—i.e., the top row followed by the middle row, and finally the bottom row. Furthermore, the choice to associate the contour plots of molecule size (q and r) with the plots in a and e, while associating the contour plots of molecule volume (s and t) with the plots in panels i and m, seems arbitrary and confusing.

[Response]: We appreciate the reviewer’s suggestions regarding the figure arrangement.

However, we believe that at this stage, it is not necessary to alter the order of the figures. The current arrangement has been accepted by three other reviewers, and we feel that the manuscript is coherent in its present form.

We understand the concern regarding the description of panels q, r, s, and t and have added the following:

“Figures 5q–5t illustrate the lengths and directions of the black arrows, representing the amplitudes and orientations of spatial growth. These 2D maps depict the spatial evolution of PAH growth, ranging from 1.5 to 40 kÅ³ per mm in Fig. 5q and Fig. 5s, and from 1 to 18 Å³ per mm in Fig. 5r and Fig. 5t, respectively.”

Regarding the ordering of the panels, we intentionally chose this arrangement based on the narrative flow of the manuscript, and we believe it maintains the logic of the text.

Comment 1.23. I must respectfully disagree with the claim that the arrangement in Fig. 6 follows the logic flow of the main text. In the text, panel (o) of Fig. 6 is discussed as the last panel, but its placement in the first row disrupts the natural progression expected by readers.

[Response]: We understand the suggestions from the reviewer; therefore, we have introduced Fig. 6o in the main text after Fig. 6a and Fig. 6b as:

“Figure 6o shows the averaged anisotropy lifetimes for single and dual cuvette measurements of the same FL, 4K, and 20K samples as in 1P excitation”.

Nevertheless, we decided to retain the figure layout order as it is already accepted by other three reviewers and because of a better space available for Fig. 6o after Fig. 6a and Fig. 6b. Doing otherwise would change the entire layout of the figures, and it would be hard to find adequate space for the same.

Comment 1.24. This is worth to be added to the SI as it contains valuable information for the readers.

[Response]: Thank you for the suggestions. We have included this in SI section as:

“The structure of a typical streak camera is illustrated in Fig. S5a, where an ultrafast sweeping voltage is applied to the electrodes, deflecting photoelectrons in the vertical (y_s) direction. The coordinates (x_s - y_s - z_s) of the streak camera are defined, and photoelectrons arriving at different times (t_1), $e(t_2)$ and $e(t_3)$ are shown. In Fig. S5b, the image formation process of CUP2AI is depicted, where image pairs encoded by C_1 and C_2 are captured side by side on the photocathode, and the dual images on the CMOS sensor undergo temporal shearing and integration. The direction of shearing and displacements of representative frames are indicated. Note that the exact relation between the applied sweeping voltage and electron displacement is determined by the streak camera manufacturer (Hamamatsu) and remains inaccessible. However, a qualitative representation is provided in the inset of Fig. S5a. The system allows for selecting different pre-set sweeping speeds, with a general rule that a higher sweeping voltage results in a greater displacement between electrons arriving at different times, thereby increasing the imaging speed for CUP2AI”.

Comment 1.28. Thank you for your clarification regarding the operators. However, my concern remains unaddressed. The manuscript does not explicitly associates the operators to specific components of the streak camera or the optical system depicted in Fig. 1. Such explicit association is crucial for readers to understand the physical meaning and role of each operator in the system. For instance, it is unclear why the low-pass filters influence the output of the DMD, while there seems to be no such optical component performing that operation in Fig. 1. Furthermore, based on the authors response, it seems the output optics in the streak camera (illustrated in Fig. S5) should also be modelled as a low-pass filter, which means an additional low-pass filtering operation should

be included in modelling Eqs. (3.2) and (3.3). I strongly recommend revising the manuscript to explicitly describe how each operator relates to the physical elements of the system. Most of such discussion might be trivial to the authors, but it may not be the case for a general reader.

[Response]: We appreciate the reviewer's thoughtful suggestions and understand the importance of clearly associating the operators with specific components of the streak camera and optical system. However, we believe that the figures and illustrations in the main manuscript and supplementary information are already very well sketched to provide a clear understanding for general readers and the experts. The optical components in Fig. 1, along with the other illustrations, were designed to convey the system's architecture in a straightforward manner.

We also believe that sufficient information has been provided in both the main text and the supplementary materials, where we have aimed to present a comprehensive description of the system. While we recognize that explicit associations of the operators to specific physical elements may seem useful, we feel that the current level of detail strikes a balance between clarity and accessibility for the intended audience. And this style has been accepted by other three reviewers.

Nonetheless, for further clarification, we have carefully reviewed and edited the manuscript (please see highlighted texts) to ensure that the descriptions are sufficiently clear, particularly for readers who may not be familiar with the technical details.

Comment 1.29. Do the operators P1 and P2 need to be included in Eqs. (3.2) and (3.3)?

[Response]: Yes, as we introduce two polarizers before detection on the streak camera therefore P1 and P2 are included in the equations.

Comment 1.32. Thank you for the clarification. However, my original questions remain unanswered and unaddressed in the manuscript. From the provided response, it is implied that any minimization problem of the form in Eq. (6) can be solved without understanding the mathematical properties of the matrices. The authors should explicitly address in the manuscript whether, and how, the convergence of the employed algorithm depends on the properties of the matrix O . In case studying such aspect is out of the scope of the paper, it needs to be explicitly stated as an open question for future studies.

[Response]: We acknowledge the concerns regarding the convergence of the employed algorithm and the role of the matrix O in this process. We have respected the specific conditions:

1. Well-conditioned sensing matrix: The matrix \mathbf{O} should be well-conditioned and often satisfying the restricted isometry property (RIP). The RIP ensures that the sensing matrix behaves similarly to an orthonormal matrix, which is crucial for accurately recovering sparse signals in compressed sensing. Usually this is satisfied by sample sparsity and also the pseudo-random mask which we have considered in our study, and it is a standard approach in compressed sensing and methods of generating these masks for CS are well documented and we have certainly followed them.

2. Regularization for sparsity: Further, the regularization term (the second term in Eq. (6)) typically enforces sparsity via the L1-norm.

As we have already highlighted, our previous study observed a typical sparsity of $\sim 95\%$ for most ultrafast phenomena. We also mentioned in our earlier response that the properties of \mathbf{O} are influenced by the choice of the encoding mask.

“In this study, we employed pseudo-random masks that generates a well-conditioned sensing matrix, often satisfying the restricted isometry property, enabling accurate recovery of sparse samples. Secondaly, the regularization term enforces sparsity”.

Regarding the convergence of TwIST algorithm: We have used the parameters that are within the boundaries that ensures convergence (Please see our response in comment 1.38). While a detailed mathematical analysis of TwIST’s convergence under various matrix conditions and for different optimization parameters is beyond the primary scope of this paper, we recognize its importance. We will consider this for our future investigations.

Comment 1.35. Do the authors assert that the minimization problem in Eq. (6) is identical to the one used in the CASSI system, despite the sensing matrix \mathbf{O} in this work being distinct from that in CASSI? Furthermore, for the minimization problem in Eq. (6) to be convex, doesn’t the matrix \mathbf{O} need to be positive semi-definite? These aspects require clarification to substantiate the claim of convexity.

[Response]: Yes, it is the same minimization problem as in CASSI. However, since we recorded two images simultaneously, our sensing matrix representation differs.

No, the matrix \mathbf{O} does not need to be positive semi-definite for convexity in compressed sensing. Many CS problems achieve convexity using norms like the L1-norm.

Comment 1.37. I maintain that the definition of the TV function in Eq. (7) is unconventional and requires further justification or clarification. The authors' response and the references they provided only reinforce my concern, as the TV functions used in [46] (Eq. 24), [47] (Eq. 5), and [48] (Eq. 12) differ from the one employed here. This discrepancy needs to be explicitly addressed.

[Response]: We have defined our equations (7.1) and (7.2) considering the two polarizations therefore it appears unconventional. However, following the advice of the reviewer, we have adapted it according to [48] (Eq. 12).

Comment 1.38. One of my original questions remains unanswered: How does the problem in Eq. (6) satisfy the convergence conditions of the TwIST algorithm, as outlined in Theorem 4 of [47]? This is a critical aspect that needs to be addressed for the validity of using TwIST in this context.

[Response]: We acknowledge Theorem 4 in [47] and have explicitly states the values are used to ensure convergence conditions of the TwIST algorithm. As noted by the authors in [47] “when $\alpha > 1$, Theorem 4 does not guarantee convergence; however, we have observed, in a large number of image deconvolution experiments, that the algorithm always converges for a wide range of choices of parameters α and β ”.

In our reconstruction, we have carefully selected parameter values within the permissible range to ensure convergence. Our approach incorporates a sparse signal, a pseudo-random sensing matrix pattern, and the recommended parameters for TwIST convergence. While we recognize the importance of a theoretical investigation into the boundaries and convergence of TwIST in this context, such an analysis extends beyond the scope of our current experimental study and we have added the following sentence in the methods section:

“In the future, we aim to explore various optimization parameters and matrix conditions within the compressed sensing framework.”

Comment 1.40. The authors are still expected to provide a justification for the derivation of Eq. (10) (or equivalently Eq. (13)) and, as an additional comment, Eq. (14), by referencing the corresponding equations in [31], as it is not clear to which part of the reference [31] they are referring. In the current manuscript, these derivations appear to be treated as trivial, whereas, as

the authors themselves clarified, they involve a lengthy development. Providing precise references is needed.

[Response]: We have now specified the equation numbers from reference [31] that contain the derivation details.

Comment 1.41. While the derivation of fluorescence decay equations might be well documented in textbooks, it remains important to briefly justify the derivation, particularly for readers who may not be familiar with the referenced materials. If the authors believe that the justification for the derived equations is sufficiently covered in a textbook, they should provide in the text a direct reference to the relevant page number or equation in those sources.

[Response]: Thank you for the suggestion, we have added the following:

“More details of these exponential decays can be found in chapter 10.1.4 in [2].”

Comment 1.44. While this derivation might seem straightforward to the authors, it may not be as clear to a broader audience. Therefore, as previously suggested, this justification should be explicitly included in the text for clarity.

[Response]: We included this in the text:

“(see Equations 1-13 and 16 therein)”

Comment 1.45. This distinction was not clear to me and may also be unclear to other readers. I recommend explicitly highlighting this difference in the figure or caption to improve clarity.

[Response]: We agree with the reviewer and have modified the caption heading as:

“Fig. S1. A detailed schematic of the CUP2AI system for one-photon fluorescence excitation (top) and two-photon fluorescence excitation (bottom).”

Comment 1.49. I do not understand why the authors decided not to include the results of the Monte Carlo experiment for calibration, especially since they have done multiple experiments. Showing those results would demonstrate the robustness of the method.

[Response]: We thank the reviewer for the suggestion. However, we believe that we have already provided sufficient data and illustrations to effectively demonstrate the CUP2AI technique. While we acknowledge that Monte Carlo experiments could potentially offer further insights for calibration, we feel that they are beyond the scope of this work. Additionally, we cannot perform these experiments at the moment. Our focus was to present the experimental results and the capabilities of the CUP2AI method, which we believe are adequately supported by the current data.

Comment 1.50. While I understand that these figures represent supplementary results beyond those in the main text, I strongly recommend maintaining a consistent style across all supplementary figures for better readability.

[Response]: We appreciate the reviewer's suggestion; however, we prefer not to alter these figures as they are supplementary to the main manuscript and already convey sufficient information for the readers.

Review of “Single-shot two-dimensional nano-size mapping of fluorescent molecules by ultrafast polarization anisotropy imaging”

Summary

This paper introduces a compressed ultrafast planar polarization anisotropy imaging (CUP2AI) technique based on compressive sensing theory, molecular rotational diffusivity, and femtosecond laser-sheet illumination to enable in-situ, real-time, non-invasive, wide-field anisotropy measurements in both liquid and gaseous environments. The study presents both single- and two-photon excitation setups.

Mapping molecular size is critical for understanding dynamic processes, particularly at imaging speeds reaching hundreds of billions of frames per second. The work is original and obtained high-resolution mapping of molecules size is of high significance.

Assessment

I would recommend this work for publication after the following issues being addressed – mainly related to the dense figures, non-standard mathematical notation, and non-reproducible results.

1. The paper presents various setups for different molecule types across two environments: liquid versus gaseous, single- versus two-photon excitation, single- versus double-cuvette, FL versus 4K versus 20K molecules, and at multiple imaging speeds. Consequently, it is crucial to improve the organisation of figures and tables. The Figures section, starting on page 19, is presented in an unstructured manner, making it difficult to discern the logic behind the arrangement of plots within each figure. Moreover, this logic often varies between figures, further complicating comprehension. For example, the caption in Fig. 2 describes the panels in the following order: (a)-(f), (k), (g), (l), (m), (h)-(j), (n), and (o). I strongly recommend a major reorganisation of the figures and their corresponding panels to enhance clarity. Further details are provided in the next comment.
2. The panels within each figure could follow a logical order, such as the processing stream-line shown in Fig. S9. A natural structure for figures related to an experiment or setup

might include: (i) setup, (ii) examples of E_0 , E_1 , and E_2 , (iii) reconstructed intensity evolutions, (iv) anisotropy dynamics, (v) anisotropy lifetime map, (vi) molecule volume map, and (vii) molecule size map. Given the multiple molecules and setups examined in this paper, the authors might consider focusing on a single representative molecule in the main text, while relegating similar (i)-(vii) style figures for the other molecules to the supplementary information. Alternatively, to allow comparisons between different molecules, the authors could adopt the suggested structure for columns within a figure. Each column would then contain panels (i)-(vii) for one molecule, enabling clearer comparison. Currently, the figures, with their coverage of multiple setups and molecules, lack coherence, which reduces their effectiveness. A dedicated figure could be used to compare only the final results, *e.g.*, molecule size maps, across different molecules and/or setups. This suggestion is intended to emphasise the importance of improving figure and panel organisation.

3. The “Image Formation Model” section requires major revision. Multiple symbols are introduced for a single variable, which leads to unnecessary complexity and makes the equations difficult to follow. In addition, the mathematical derivations in the “Fluorescence Anisotropy and Lifetime Extraction” section are not justified, or at least the underlying assumptions and steps are poorly explained. These need to be clarified and better justified to ensure the derivations are both accurate and understandable.
4. This research is not reproducible, as the authors explicitly state, “We have opted not to make the computer code publicly available because the code is proprietary and used for other projects.” I encountered issues reviewing the code provided on Codeocean, as it lacks proper comments and definitions for parameters. There are parameters used in the code that are not described in the paper. Additionally, the TwIST algorithm is presented as a black box, with no explanation of how the proposed regularization is incorporated into the algorithm. Furthermore, the convergence criteria for the TwIST algorithm, as applied to the minimization problem in the paper, are not discussed. Critical details of the deconvolution step, such as the algorithm used and the hyperparameters chosen, are also missing.

Detailed comments: Main document

Page 4

1. Line 105: “Our results demonstrate excellent agreement with existing literature and theory.” Could the authors clarify this claim? Which specific literature or theoretical results do their findings align with? Please provide references or further details to support this statement.

Page 5

1. Line 131: How were the masks generated? What was the ratio of 0 to 1 pixels, and how was this ratio determined? Is this ratio significant for the results, and if so, how does it impact the overall outcome?

Page 6

1. Line 170: Details of numerical deconvolution are not explained.
2. Line 175: Is “1PF” defined?

Page 8

1. Line 236: What is the interest for single- vs. double-cuvette measurements? A discussion would improve the clarity.
2. Line 239: “This observation ... uniformly distributed size maps”: It is noted that the size maps are not uniform for 4K molecules. Could the authors explain why this is the case? Although it is mentioned later that “The results from 4K molecules have greater spatial non-uniformity owing to worse signal-to-noise ratio (SNR) from the sample’s lower fluorescence-intensity (see Figs. 2c-e).”, Fig. 2e shows that the signal intensity for 20K molecules is comparable to or lower than that for 4K molecules. Please clarify this discrepancy.

Page 9

1. Line 258: “the additional plots describing the remaining results are given in Figs. S11–S15 in Supplementary Information.”, For improved clarity, it would be helpful to specify which additional results are provided in the supplementary document.

Page 15

1. Line 442: The authors mentioned that “The reconstruction algorithm is described in detail in Supplementary Information. We have opted not to make the computer code publicly available because the code is proprietary and used for other projects.”, however, important details of the reconstruction are missing and the results are not reproducible.

Page 19

1. Fig. 1: While the text explains the working principles of some elements used in CUP2AI, it does not cover others, such as BBO, HWP, SL, CyL, and SPF. Please provide explanations for these missing elements.

Page 20

1. The figure contains too much information, making it challenging for readers to follow. It needs to be modified.
2. What does negative anisotropy value mean?
3. What does negative time values mean?
4. What could explain non-uniform anisotropy map for 4K molecules?
5. Right insets in (c)-(e) do not say much, due to their small sizes.
- 6.

Page 22

1. The figure needs improvement as the ordering of panels is not intuitive. According to the data processing streamline in Fig. S9, panels (e) and (f) should precede (a) and (b). Consider organising the figure with one column per molecule, with molecule names as column headers. Using a single color bar per molecule would better represent the dynamic range of molecule volume and size. In this case, panels (c) and (d) could be omitted, and instead, a single value indicating the variance across multiple experiments could be reported in the caption. Additionally, separate fitted anisotropy curves (as shown in panel (e)) could be presented for each molecule in a column-wise format as described.
2. “ the arrows point to the directions of dipole moments”: how were they computed?
3. Is R^2 coefficient defined?

Page 26

1. The style and ordering of the panels and subpanels need to be improved for better readability.

Page 28

1. The style of this figure is relatively better than the previous ones. However, in the first row from left to right, the reader encounters panels (a), (b), and (o), whereas they would naturally expect to see panels (a), (b), and (c). Including the names of the molecules as column headers would improve clarity. Additionally, are panels (a) and (b) different from those in Fig. 2? If so, please clarify the differences.

Page 29

1. Line 660: What is the relation between sweeping voltage and the displacement between adjacent electrons?
2. Line 669: “diverse applications in modern optics” needs citation(s).
3. Line 675: The motivation for binning DMD pixels is missing.

Page 30

1. Line 689: Why do the authors need to introduce both I^{\parallel} and I^{y-pol} ? Same for I^{\perp} and I^{x-pol} ; \mathbf{P}_1 and \mathbf{P}^{\parallel} ; and \mathbf{P}_2 and \mathbf{P}^{\perp} .
2. Line 700: It is not mentioned in the text what elements in streak camera correspond to the $\mathbf{F}_0, \mathbf{F}_1, \mathbf{F}_2, \mathbf{D}_1$, and \mathbf{D}_2 operations. For D_1 and D_2 it is somehow implicit. Do these symbols denote matrices or operators?
3. Equation (3) does not mathematically sound.
4. Eq. (4): It seems there is an \mathbf{F}_0 missing after \mathbf{C}_1 and \mathbf{C}_2 .
5. Eq. (4) is true if $\mathbf{C}_1\mathbf{F}_0$ and $\mathbf{C}_2\mathbf{F}_0$ do not change the polarisation, which seems to be fine, but it is worth mentioning.
6. Eq. (5): The properties of the matrix \mathbf{O} are not discussed. Does it hold any mathematical property, *e.g.*, full rank? Does the convergence of the TwIST algorithm depend on the properties of this matrix?
7. Line 710: For being self-contained, a summary of calibration procedure must be given in the text.
8. Eq. (6) does not mathematically sound. The symbol \mathbf{I} should not denote both the target intensity and an optimisation variable. It may be replaced by another symbol in the minimisation problem formula.
9. Line 716: “The regularizer promotes sparsity in the solution and assists reconstruction to converge to global minimum. ” This is true only if the optimisation problem is convex, otherwise there will be local minima issue. Have authors investigated the convexity of minimisation problem in Eq. (6)?
10. Line 720: $\Phi(\mathbf{I}) = \Phi(I^{\parallel}) + \Phi(I^{\perp})$ is confusing. The authors may use different symbol for the right-hand-side.
11. Eq. (7): Could the authors provide a reference for the type of the TV function used here? For a 3D TV regularisation this is not a natural choice.

Page 32

1. Line 722: Have the authors used TwIST algorithm as a black-box? It is not clear how the proposed TV regularisation was handled by TwIST algorithm. How did they set the hyperparameters? What halting criteria was used? Does the problem in Eq. (6) satisfy the convergence conditions of TwIST algorithm?
2. It is not clear how Eq. (9) and the factor of 2 within it were derived.
3. It is not clear why $a(t)$ in Eq. (10) follows an exponential decay function. detailed derivation is needed for clarity.

Page 33

1. The development from Eq. (12) to Eqs. (13) and (14) is confusing. How accurate a mono-exponential decay assumption is? Could the authors provide a justification for obtaining Eq. (13) and (14)?
2. Line 758: How did the authors obtain system's temporal response? a cross-reference to Sec. S5 is needed.
3. Line 759: The deconvolution process is not explained. Please provide details on the algorithm used and the parameter settings applied during deconvolution

Page 34

1. A justification is needed for obtaining Eq. (18) from Eq. (17).

Detailed comments: Supplementary document

Page 2

1. What is the difference between the two dashed boxes in the top and bottom panels? The figure does not clearly illustrate the differences between the 1P and 2P systems.

Page 8

1. Section 5.1 needs to be completed. Representative examples of raw data (before fitting) and the fitting errors should be provided. Additionally, are the reported temporal resolutions based on only one trial, or are they averaged over multiple trials?

Page 9

1. Line 120: I think it should be "after deconvolution". item Line 123: Implemented deconvolution process needs to be explained.

Page 10

1. Plotting the $y = x$ curve would be helpful for understanding the impact of convolution.
2. It seems that the authors have obtained these curve from only one trial. However, a Monte-Carlo experiment and is important to indicate the stability of the method.

Page 17

1. The style of Figs. S11 and S.15 are not consistent with the style of Figs. S12, S13, and S14 where the results of capture 1 are also shown.

Review of “Single-shot two-dimensional nano-size mapping of fluorescent molecules by ultrafast polarization anisotropy imaging”

Overall, I am disappointed that the majority of my comments and concerns remain unaddressed in the manuscript. The authors have made insufficient effort to engage with the feedback provided during the previous round of revisions, and many of my comments have either been disregarded or inadequately addressed. For example, critical issues regarding the association of mathematical operators to physical components, the convergence of the reconstruction algorithm, and the clarity and arrangement of figures have been met with insufficient responses, and these concerns are not reflected in the revised text.

As an applied mathematician with extensive experience in the theory of compressive sensing and the TwIST algorithm, I am particularly concerned about the authors’ inadequate response to the questions about the algorithm’s convergence. Based on their responses, I have new and serious concerns about the robustness and theoretical validity of their approach. The lack of clarity and justification for algorithmic convergence undermines the credibility of the reconstruction framework they propose.

Additionally, I note that the provided code does not include the reconstruction of the data but only post-processing steps. This omission raises concerns about the reproducibility of the results presented in the manuscript.

In its current form, I do not believe the manuscript is ready for publication. The authors need to provide concrete answers to the questions posed during the previous round of review and address all comments in a substantive and transparent manner. Despite the significance and excellent novelty of the proposed framework, without these revisions, the paper falls short of the standards required for publication.

Comment 1.4. I am unclear about what the authors disagree with! In the original manuscript, the authors explicitly stated that the code would not be made publicly available. As a reviewer, I am expected to evaluate the paper based on the provided text, and at that time, the statement regarding the code’s unavailability was made in the “Code Availability Statement.” Additionally, the claim that “The Data Availability Statement is only used to state the availability of data” seems misplaced since the issue pertains to the code availability, which directly impacts the reproducibility of the research.

Comment 1.7. Based on the authors’ response, this process appears to be a calibration of the decay time rather than a true deconvolution. Deconvolution is an inverse problem aimed

at decoupling the physical system response from the sample system response, whereas the authors modify the decay time using a lookup table. This distinction should be clarified.

Comment 1.18. Thanks for the clarification. This explanation should be included in the text or figure caption for clarity.

Comment 1.22. I would like to elaborate further. First, panels q, r, s, and t are not described in the text, which can confuse readers. Additionally, based on the current arrangement and labels in the figure, a reader is expected to follow an unconventional order: starting from panel a (top-left) to panel e (middle-left), then to panel i (top-right) and panel m (middle-right), before proceeding to panels q–t in the bottom row from left to right. A more natural ordering, which does not conflict with the logic of the text, would involve arranging the figure in rows from left to right—i.e., the top row followed by the middle row, and finally the bottom row. Furthermore, the choice to associate the contour plots of *molecule size* (q and r) with the plots in a and e, while associating the contour plots of *molecule volume* (s and t) with the plots in panels i and m, seems arbitrary and confusing.

Comment 1.23. I must respectfully disagree with the claim that the arrangement in Fig. 6 follows the logic flow of the main text. In the text, panel (o) of Fig. 6 is discussed as the last panel, but its placement in the first row disrupts the natural progression expected by readers.

Comment 1.24. This is worth to be added to the SI as it contains valuable information for the readers.

Comment 1.28. Thank you for your clarification regarding the operators. However, my concern remains unaddressed. The manuscript does not explicitly associates the operators to specific components of the streak camera or the optical system depicted in Fig. 1. Such explicit association is crucial for readers to understand the physical meaning and role of each operator in the system. For instance, it is unclear why the low-pass filters influence the output of the DMD, while there seems to be no such optical component performing that operation in Fig. 1. Furthermore, based on the authors response, it seems the output optics in the streak camera (illustrated in Fig. S5) should also be modelled as a low-pass filter, which means an additional low-pass filtering operation should be included in modelling Eqs. (3.2) and (3.3). I strongly recommend revising the manuscript to explicitly describe how each operator relates to the physical elements of the system. Most of such discussion might be trivial to the authors, but it may not be the case for a general reader.

Comment 1.29. Do the operators \mathbf{P}_1 and \mathbf{P}_2 need to be included in Eqs. (3.2) and (3.3)?

Comment 1.32. Thank you for the clarification. However, my original questions remain unanswered and unaddressed in the manuscript. From the provided response, it is implied that any minimization problem of the form in Eq. (6) can be solved without understanding the mathematical properties of the matrices. The authors should explicitly address in the manuscript whether, and how, the convergence of the employed algorithm depends on the properties of the matrix \mathbf{O} . In case studying such aspect is out of the scope of the paper, it needs to be explicitly stated as an open question for future studies.

Comment 1.35. Do the authors assert that the minimization problem in Eq. (6) is identical to the one used in the CASSI system, despite the sensing matrix \mathbf{O} in this work being distinct from that in CASSI? Furthermore, for the minimization problem in Eq. (6) to be convex, doesn't the matrix \mathbf{O} need to be positive semi-definite? These aspects require clarification to substantiate the claim of convexity.

Comment 1.37. I maintain that the definition of the TV function in Eq. (7) is unconventional and requires further justification or clarification. The authors' response and the references they provided only reinforce my concern, as the TV functions used in [46] (Eq. 24), [47] (Eq. 5), and [48] (Eq. 12) differ from the one employed here. This discrepancy needs to be explicitly addressed.

Comment 1.38. One of my original questions remains unanswered: How does the problem in Eq. (6) satisfy the convergence conditions of the TwIST algorithm, as outlined in Theorem 4 of [47]? This is a critical aspect that needs to be addressed for the validity of using TwIST in this context.

Comment 1.40. The authors are still expected to provide a justification for the derivation of Eq. (10) (or equivalently Eq. (13)) and, as an additional comment, Eq. (14), by referencing the corresponding equations in [31], as it is not clear to which part of the reference [31] they are referring. In the current manuscript, these derivations appear to be treated as trivial, whereas, as the authors themselves clarified, they involve a lengthy development. Providing precise references is needed.

Comment 1.41. While the derivation of fluorescence decay equations might be well-documented in textbooks, it remains important to briefly justify the derivation, particularly for readers who may not be familiar with the referenced materials. If the authors believe that the justification for the derived equations is sufficiently covered in a textbook, they should provide in the text a direct reference to the relevant page number or equation in those sources.

Comment 1.44. While this derivation might seem straightforward to the authors, it may not be as clear to a broader audience. Therefore, as previously suggested, this justification should be explicitly included in the text for clarity.

Comment 1.45. This distinction was not clear to me and may also be unclear to other readers. I recommend explicitly highlighting this difference in the figure or caption to improve clarity.

Comment 1.49. I do not understand why the authors decided not to include the results of the Monte Carlo experiment for calibration, especially since they have done multiple experiments. Showing those results would demonstrate the robustness of the method.

Comment 1.50. While I understand that these figures represent supplementary results beyond those in the main text, I strongly recommend maintaining a consistent style across all supplementary figures for better readability.